



# The historical ozone trends simulated with the SOCOLv4 and their comparison with observations and reanalysis

Arseniy Karagodin-Doyennel[1,2,*], Eugene Rozanov[1,2,3,*], Timofei Sukhodolov[1,2,4,*], Tatiana Egorova[1], Jan Sedlacek[1], William Ball[5,+], and Thomas Peter[2]

[1]Physikalisch-Meteorologisches Observatorium Davos/World Radiation Center (PMOD/WRC), Davos, Switzerland
[2]Institute for Atmospheric and Climate Science (IAC), ETH, Zurich, Switzerland
[3]Saint Petersburg State University, Saint Petersburg, Russia
[4]Institute of Meteorology and Climatology, University of Natural Resources and Life Sciences, Vienna, Austria
[5]Department of Geoscience and Remote Sensing, TU Delft, Delft, Netherlands
[*]These authors contributed equally to this work.
[+]Deceased.

**Correspondence:** Arseniy Karagodin-Doyennel (darseni@student.ethz.ch)

**Abstract.**

There is evidence that the ozone layer has begun to recover owing to the ban on the production of halogen-containing ozone-depleting substances (hODS) accomplished by the Montreal Protocol and its Amendments (MPA). However, recent studies, while reporting an increase in tropospheric ozone and confirming the ozone recovery in the upper stratosphere, also indicate

a continuing decline in the lower tropical and mid-latitudinal stratospheric ozone. While these are indications derived from observations, they are not reproduced by current global chemistry-climate models (CCMs), which show positive or near-zero trends for ozone in the lower stratosphere. This makes it difficult to robustly establish ozone recovery and has sparked debate about the ability of contemporary CCMs to simulate future ozone trends. We applied the new Earth system model SOCOLv4 to calculate long-term ozone trends and compare them with trends derived from observations and reanalyses. The analysis is

performed separately for the ozone depletion [1985-1997] and the ozone recovery [1998-2018] periods. Within the 1998-2018 period, SOCOLv4 shows clear ozone recovery in the mesosphere, upper and middle stratosphere; no significant ozone trend in the extra-polar lower stratosphere; and a steady increase in the tropospheric ozone. However, the lower stratospheric ozone trends remain controversial because the reanalysis datasets and SOCOLv4 results suggest slightly negative but insignificant trends which do not agree with some observation composite analysis. The obtained pattern of ozone trends is in general

agreement with observations and reanalysis data sets, confirming that modern chemistry-climate models such as SOCOLv4 are generally capable of simulating the observed ozone changes, justifying their use to project the future evolution of the ozone layer.

## 1   Introduction

There is evidence that the restrictions on halogenated ozone-depleting substances (hODS) introduced by Montreal Protocol

and its Amendments (MPA) are beginning to take effect, leading to observable ozone recovery at certain latitudes and altitudes



(WMO, 2014, 2018; McKenzie et al., 2019). The positive role of MPA has also been confirmed by model projection of ozone depletion using the "world avoided" scenario (Newman et al., 2009; Garcia et al., 2012; Egorova et al., 2013; Goyal et al., 2019). Using the future simulation of multiple CCMs, Dhomse et al. (2018) estimated the approximate time of the ozone recovery back to the pre-1980 level for different regions, again highlighting the success of MPA in protecting ozone.

In the mid-1990s, hODS mixing ratios reached their maximum, and thereafter started dropping again by virtue of the MPA. The annual mean total column ozone in the mid-1990s is quantified to be about 5% below its pre-1980 level globally, and about 17% lower over Antarctica (WMO, 2018). Starting with the turn of the century, total ozone is expected to recover, but given the long lifetimes of some hODS and the simultaneously changing climate, it is not easy to project how long it will take for ozone to return to the pre-1980 level. The near-global total column ozone—stratospheric plus tropospheric—remained

stable since 2000, showing no or a slightly positive trend, albeit statistically not significant (WMO, 2014, 2018; Ball et al., 2018). It should be noted that not all greenhouse gasses (like $CO_2$ or $CH_4$) contribute positively to ozone recovery and, for instance, $N_2O$ is expected to be the most important anthropogenic compound that will slow down ozone recovery in the future (Chipperfield, 2009). Nonetheless, even super recovery (when the total column ozone exceeds pre-ozone hole level) over the mid-latitudes imposed on greenhouse gasses is expected (WMO, 2010). The severe Antarctic ozone depletion, referred to as

the "ozone hole", also started slowly to recover (Salby et al., 2011; Solomon et al., 2016).

Although the evolution of the total column ozone is well studied, the evolution of ozone in different atmospheric layers has its own characteristics.

In the troposphere, the ozone concentration has been continuously increasing (Mickley et al., 2001) due to the continuous increase in tropospheric ozone precursors (Griffiths et al., 2021). Hence, a positive trend in tropospheric ozone is expected

(Ziemke et al., 2019). At the same time, simulating the evolution of tropospheric ozone remains challenging because its changes depend on complex, interdependent interactions among precursor emissions, atmospheric transport, photochemical production, deposition, and troposphere-stratosphere exchange. Recent results indicate that models cannot properly capture the tropospheric ozone trends, and that tropospheric ozone variability diverges widely among models (Parrish et al., 2014; Revell et al., 2018; Zhang and Cui, 2022).

In the upper stratosphere (above 10 hPa), the ozone abundance declined by about 4-8% between 1980 to the late 1990s due to a continuous increase in stratospheric chlorine loading (e.g., Steinbrecht et al., 2017; WMO, 2018), but since the late 1990s it has been steadily recovering (Ball et al., 2018; WMO, 2018). This recovery is statistically robust according to various analyses of satellite observations and model simulations (Harris et al., 2015; Steinbrecht et al., 2017; Ball et al., 2018; Godin-Beekmann et al., 2022). The main driver of the upper stratospheric ozone recovery is the reduction of halogen loading. In addition, the tem-

perature decrease caused by increases in greenhouse gasses (GHGs), primarily $CO_2$, slows down the temperature-dependent catalytic ozone loss cycles (Hitchman and Brasseur, 1988; Zubov et al., 2013; WMO, 2018). Via the reaction $CH_4 + Cl \rightarrow CH_3 + HCl$, the increase in $CH_4$ further promotes the recovery of the ozone layer in the upper stratosphere (Hitchman and Brasseur, 1988).

Lower stratospheric ozone (LSO) has recovered more slowly than expected, if at all (Ball et al., 2018, 2019). Ball et al. (2018)

based their analysis on dynamical linear regression modeling (DLM), a class of non-stationary time series models, which are





the next generation of multiple linear regression (MLR) that considers the level of trend non-linearity, thereby increasing the accuracy of the estimate. Ball et al. (2018) applied DLM to the homogenized BAyeSian Integrated and Consolidated (BASIC) composite ozone time series, which was derived from satellite observations (Ball et al., 2017). Using partial ozone columns, a statistically robust negative tendency was found for the layer between 146 and 32 hPa on the near-global scale (55°N-55°S).

Ball et al. (2018) reasoned that the detection of this negative trend in LSO was previously prevented by not properly excluding the contribution from increasing tropospheric ozone, which partially compensates for the negative LSO changes in TCO analyses (Ziemke et al., 2019). The results of Ball et al. (2018) have sparked intense debate in the scientific community regarding the nature of the decline in extratropical LSO, and also expressed doubts about its existence. Chipperfield et al. (2018) found a strong increase in LSO in 2017 by about 60% in the Southern Hemisphere, prompting them to suggest that large natural inter-annual variability might be behind the apparent negative tendencies in LSO. However, in a subsequent study, Ball et al. (2019) have analyzed the extended ozone time series, showing that the short-term LSO increase in 2017 could be traced back to changes in the quasi-biennial oscillation (QBO), but that the negative trend in LSO continued to be observed after 2017. This indicates that the LSO is still poorly understood in terms of its long-term variability.

Besides a change in the relative strengths of the lower and upper branches of the BDC there might be additional reasons for the decline in LSO, including: the recent reduction in solar activity (Arsenovic et al., 2018); the influence of halogen-containing very short-lived species and other gases unaccounted for by the MPA (Hossaini et al., 2015; Oman et al., 2016; Oram et al., 2017); increased emissions of inorganic iodine (Cuevas et al., 2018; Karagodin-Doyennel et al., 2021); increased aerosol loading (Andersson et al., 2015); insufficient treatment of diffusion and transport processes in models (Dietmüller et al., 2017, 2018); unexpected increase in emissions of CFC-11 violating the MPA (Fleming et al., 2020); altitude changes in the extratropical tropopause (Bognar et al., 2022).

Recently, the ozone simulated by 31 CCMs participating in the Chemistry-Climate Model Initiative Phase-1 (CCMI-1) has been analyzed, and it was shown that the pattern of the signal in LSO varies greatly even between different realizations of the experiment performed with the same CCM (Dietmüller et al., 2021). The question is why models cannot reproduce the persistent and statistically robust declining trends in near-global LSO.

Stone et al. (2018) investigated the impact of changes in atmospheric dynamics on the stratospheric ozone trends. To this end, they performed linear regression analysis on a nine-member ensemble of free-running and nudged simulations with the Whole Atmosphere Community Climate Model, version 4 (WACCM4). They conclude that the large dynamic variability in the extratropical lower stratosphere prevents the detection of a stable ozone trend. These results suggest that to study long-term ozone trends in dynamically controlled regions, it is desirable to analyze different ensemble members of the experiment separately because ensemble-averaging suppresses the variability. In contrast, Dietmüller et al. (2021) suspect that the models have difficulties reproducing the observed LSO trend because of extreme natural variability at the beginning or end of the observed period (1998–2018), or because natural variability in QBO and BDC is not adequately simulated. Thus, small inadequacies in atmospheric dynamics might be the reason why models do not demonstrate robust LSO negative tendencies, amplified by the strong internal ozone variability (Shangguan et al., 2019). However, even when using models with assimilated dynamics (i.e., specified dynamics or nudged simulations), the negative tendencies in LSO are still not revealed in model simulations (Ball





et al., 2018). At this point, it should be mentioned that reanalysis data used for nudging in CCMs can introduce noise, e.g., in the vertical fluxes, which may impact model performance (Chrysanthou et al., 2019).

Therefore, it is still unclear whether LSO will reach the 1980-level in the future. An intensification of ozone-poor air transport from the troposphere related to an acceleration of the meridional Brewer-Dobson circulation (BDC, Butchart et al., 2006)

between 1998 and 2016 was suggested as the primary mechanism for the declining ozone trend in tropical LSO (Zubov et al., 2013; Wargan et al., 2018; Orbe et al., 2020). However, this mechanism cannot explain ozone decline over the middle latitudes discovered by Ball et al. (2018) because intensified BDC should rather increase ozone concentration in this area.

Tropical LSO is projected to decline further throughout the 21st century (Zubov et al., 2013). This is caused by a warming of the tropical upper troposphere, increasing the tropical-to-extratropical temperature gradient, which in turn pushes the subtrop-

ical jet upward, lifting the tropopause and accelerating the meridional transport, especially via the shallow branch of the BDC (Zubov et al., 2013). But the future evolution of the middle latitudes LSO is not clear. This problem casts some doubts on the applicability of the state-of-the-art chemistry-climate models to project the future evolution of the ozone layer. Addressing this issue requires more simulations and careful analysis of the historical ozone trends, and thus motivates the present study.

Here, we aim to evaluate ozone trends from 1985-2018 from the ground to the mesosphere using the Earth System Model

(ESM) SOCOLv4 (Sukhodolov et al., 2021). To verify if the SOCOLv4 can reproduce the statistically reliable observed ozone trends, we applied DLM to SOCOLv4 simulations, the BASIC composite, and several re-analysis datasets. Section 2 introduces the SOCOLv4 model, the ensemble reference experiment setup, as well as datasets used in our study. Section 3 outlines the dynamic linear approach employed in this study to derive the trends. Section 4 provides the results, starting with the DLM analysis of trends in reactive species and temperature from SOCOLv4 and continuing with the comparison of ozone trends

from SOCOLv4 to those from the BASIC observational composite and from the reanalysis data. A discussion of the results and general conclusions are presented in Section 5.

## 2 SOCOLv4 chemistry-climate model, experiment setup, and data description

The Earth System Model (ESM) SOCOLv4 (SOlar Climate Ozone Links version 4) is based on the Max Planck Institute for Meteorology (MPI-M) Earth System Model (ESM) version 1.2 (MPI-ESM1.2) (Mauritsen et al., 2019) that is interac-

tively coupled to the chemical module MEZON (Model for Evaluation of oZONe trends) (Rozanov et al., 1999; Egorova et al., 2003) and the size-resolving sulfate aerosol microphysical module AER (Weisenstein et al., 1997; Sheng et al., 2015; Feinberg et al., 2019). MPI-ESM1.2 consists of the sixth generation of the Atmospheric General Circulation Model (AGCM) ECHAM6 (the Middle Atmosphere version of the European Centre/Hamburg Model 6), the Max-Planck-Institute for Meteorology Ocean Model (MPIOM), the Hamburg Ocean Carbon Cycle (HAMOCC) model simulating the ocean biogeochemistry and

the Jena Scheme for Biosphere-Atmosphere Coupling in Hamburg (JSBACH) used as ecosystem model/land surface component. ECHAM6 is built on a spectral dynamical core and contains modules to compute radiation, convection, cloud processes, and atmospheric transport. The transport scheme is based on the flux-form semi-Lagrangian scheme (Lin and Rood, 1996). The chemical module MEZON includes roughly 100 chemical species linked by 216 gas-phase reactions, 72 photolysis re-



actions, and 16 heterogeneous reactions in/on aqueous sulfuric acid aerosols and polar stratospheric clouds (STS, NAT, ICE

types) using the implicit iterative Newton–Raphson scheme (Ozolin, 1992; Stott and Harwood, 1993). The photolysis rates are calculated using an online look-up-table approach (Rozanov et al., 1999), including effects of the solar irradiance for the spectral interval 120-700 nm. ECHAM6, MEZON, and AER are interactively coupled by the 3-dimensional meteorological fields of wind and temperature as well as by the radiative forcing of sulfate aerosol, $O_3$, $H_2O$, $CH_4$, $N_2O$, and chlorofluorocarbons (CFCs). Conventionally, the SOCOLv4 is formulated on the horizontal spectral resolution with T63 triangular truncation (96 x

192 or 1.9°x 1.9° grid spacing) and 47 vertical levels from the surface to 0.01 hPa in a sigma-pressure coordinate system. The main time step in SOCOLv4 is 15 min, whereas full radiation and chemistry calculations are carried out every 2 hours. In this study, we analyze long-term ozone time series from the SOCOLv4 ensemble reference experiment carried out using standard conditions. The runs were initiated from the MPI-ESM1.2 restart files for the year 1970, while chemistry was initiated from the SOCOLv3 runs (Revell et al., 2016). This experiment was performed for the period 1949-2018. In 1980, the experiment was

branched into six ensemble members which are initialized with slightly varying initial conditions namely a first-month small (about 0.1%) perturbation in the CO2 concentration in order to have different realizations of further ozone evolution thereby describing the internal model variability.

The boundary conditions and forcing data for the SOCOLv4 reference experiment are based on CMIP6 recommendations (Eyring et al., 2016) and given by the input datasets from the Model Intercomparison Projects (input4MIPs) database [1]. All

climate forcings are historical before 2015 and for the years 2015 to 2018 they are switched to the SSP2-4.5 scenario (O'Neill et al., 2016). A detailed description of all modules incorporated into SOCOLv4 as well as more details on the conducted reference experiment can be found in the SOCOLv4 validation paper (Sukhodolov et al., 2021). In this study, the SOCOLv4 runs are in a free-running mode except for QBO which is nudged to observations.

Here, we analyze the historical period from 1985 to 2018 which is well-covered by observations. In our analysis, we split this

period into two parts: 1985-1997 is the ozone depletion phase, and 1998-2018 is the ozone recovery phase. Also, in our study, to compare the obtained stratospheric ozone trends in SOCOLv4, we use the BASIC ozone composite. The BASIC composite is the ozone time-series dataset built from a Bayesian joint self-calibration analysis of several composite ozone datasets [2]. A detailed description of how the BASIC composite was produced can be found in Ball et al. (2017). The BASIC was specifically designed for the trend analysis, hence it is a good candidate to compare the modeled ozone trends.

For a number of atmospheric layers between the ground and the mesosphere we compare the SOCOLv4 simulations with (i) the BASIC ozone composite, (ii) the Multi Sensor Re-analysis data version 2 (MSRv2, van der A et al., 2015)[3], (iii) the Modern-Era Retrospective analysis for Research and Applications, Version 2 (MERRA-2, Gelaro et al., 2017)[4], as well as (iv) the 4D-Var data assimilation-based comprehensive European ReAnalysis (ERA-5, Hersbach et al., 2020)[5].

---

[1]More information on the input4MIPs database can be found here: https://esgf-node.llnl.gov/search/input4mips/

[2]The BASIC composite dataset can be downloaded here: https://data.mendeley.com/datasets/2mgx2xzzpk/3

[3]The MSRv2 reanalysis dataset can be found here:https://www.temis.nl/protocols/O3global.php

[4]The MERRA-2 reanalysis dataset can be found here: https://disc.gsfc.nasa.gov/datasets?project=MERRA-2

[5]The ERA-5 reanalysis dataset can be found here: https://cds.climate.copernicus.eu/!/search?text=ERA5type=dataset



## 3   The description of the DLM approach

To obtain ozone trends from the model and other datasets, we applied the regression analysis using DLM with the state-space approach described in Laine et al. (2014); Ball et al. (2018), and a tutorial on the method can be found here [6]. DLM is an advanced stochastic model for detecting long-term trends in the prognostic variables imposed by well-known processes driven by explanatory/proxy variables and unaccounted processes. The DLM consists of the slowly varying background level, seasonal components, reaction to the external forcing of well-known processes modeled by explanatory variables, and stochastic noise

allowing for residuals in a first-order autoregressive (AR1) process. Compared to simpler statistical multivariate analysis (such as multiple linear regression (MLR)), the application of DLM allows more accurate conclusions about the variability due to considering the degree of trend non-linearity.

In this work, the used proxies (see Figure A1 in appendix) of atmospheric variability are chosen following Ball et al. (2018). We use the solar forcing represented by 30 cm solar radio flux (F30, sfu) (Dudok de Wit et al., 2014), and zonal winds at 30

and 50 hPa (in m/s) which are the two principal components of the Quasi-biennial oscillation (QBO)[7] variability, a latitude-dependent stratospheric aerosol optical depth (SAOD, dimensionless) to represent volcanic activity (Thomason et al., 2018); the El Niño–Southern Oscillation variability represented by ENSO's 3.4 index (ENSO, degree K)[8], and the Arctic (AO) and Antarctic (AAO) Oscillation indexes (hPa)[9] are used to represent trends in total/partial ozone.

The above regressors (except solar forcing) are used only to analyze the ozone changes in the BASIC composite and reanalysis

datasets. In SOCOLv4, for every ensemble member, the AO, AAO, ENSO, and SAOD proxy variables are calculated directly from the simulated meteorological fields of geopotential height (at 1000 and 700 mb), sea surface temperature (SST), and aerosol extinction at 300-500 nm band. To analyze the model data, we took the QBO proxies at 25 hPa instead of at 30 hPa, because the low model resolution caused a high correlation between QBO at 30 and 50 hPa, which is inappropriate for regression analysis. The solar forcing used to derive ozone trends from SOCOLv4 is the same as for observations (Solar F30

index). The independence of regressors is a crucial point of such an analysis. To this end, we performed a correlation test for each proxy from the six ensemble members of the experiment and 2 periods of analysis. Figure 1 illustrates the results of proxies correlation analysis.

---

[6]Dynamic linear model tutorial: https://mjlaine.github.io/dlm/dlmtut.html,

[7]The data provided by Freie Universität Berlin: https://www.geo.fu-berlin.de/en/met/ag/strat/produkte/qbo/index.html

[8]The NOAA website: https://www.psl.noaa.gov/enso/mei.old/

[9]National Centers for Environmental Prediction (NCEP): /www.cpc.ncep.noaa.gov/products/precip/CWlink/



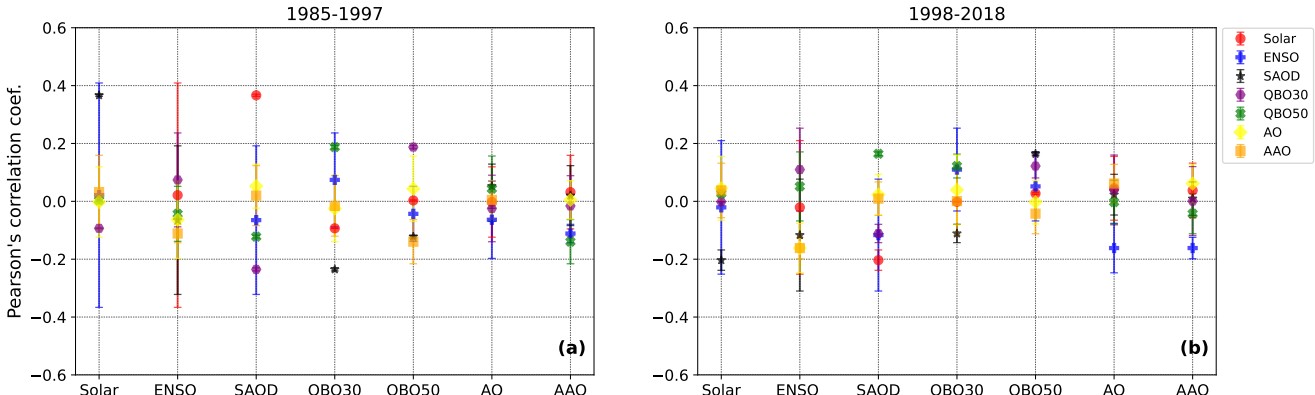

**Figure 1.** Pearson's linear correlation coefficients for different covariate variables (proxy variables) of the SOCOLv4 ensemble experiment: **(a)** for ozone depletion phase [1985-1997]; **(b)** for ozone recovery phase [1998-2018]. Error bars represent the 2-σ standard deviation of correlation coefficients between ensemble members.

The correlation test indicates generally low mutual dependencies, except for the correlation between solar radio flux and SAOD for the ozone-depleting period. The correlation between solar fluxes and volcanic activity has been investigated before (Kuchar et al., 2017). It is related to strong volcanic eruptions like Pinatubo and El Chichón that affected the atmosphere during the 1985-1997 period, which happened to synchronize with the solar cycle. The remaining proxies show the average correlation coefficients within 0.20-0.25, which is considered to be a "weak" mutual dependency. It is important to note that all DLM calculations were performed for the entire 1985-2018 period. Then, the ozone trends per decade for the intervals ([between 1985 and 1997] and [between 1998 and 2018]) are calculated from the output DLM output. We determine the background level of the model from DLM by means of Kalman filtering, which allows to remove the impact of forcings on the natural variability. We produced 200 samples from the model states accounting for the posterior uncertainty estimated by the Markov chain Monte-Carlo method (see Laine et al. (2014)) to find the mean background level component and its standard deviation between samples to calculate the statistical significance. Finally, we calculated the trend per decade for two periods from the sample-mean background level. The statistical significance is determined using the Student's test. The ensemble mean of the SOCOLv4 experiment is calculated as the average of the DLM output from all individual ensemble members.

## 4 Results

### 4.1 Simulated long-term trends in $O_3$, $ClO_x$, $BrO_x$, $NO_x$, $HO_x$, and temperature for the depletion and recovery periods

The explanation of simulated ozone trends for different atmospheric layers cannot be properly done without demonstrating the trends not only in ozone itself, but trends in reactive species involved in the ozone destruction cycles. In addition, the temperature is accounted for, because it controls the rate of chemical reactions for the formation of radicals and is an indicator





of ozone change, since ozone is a radiatively active gas. Decadal trends in ozone as well as radicals and temperature from the SOCOLv4 ensemble mean results in both the ozone depletion and ozone recovery periods are presented in Figure 2.

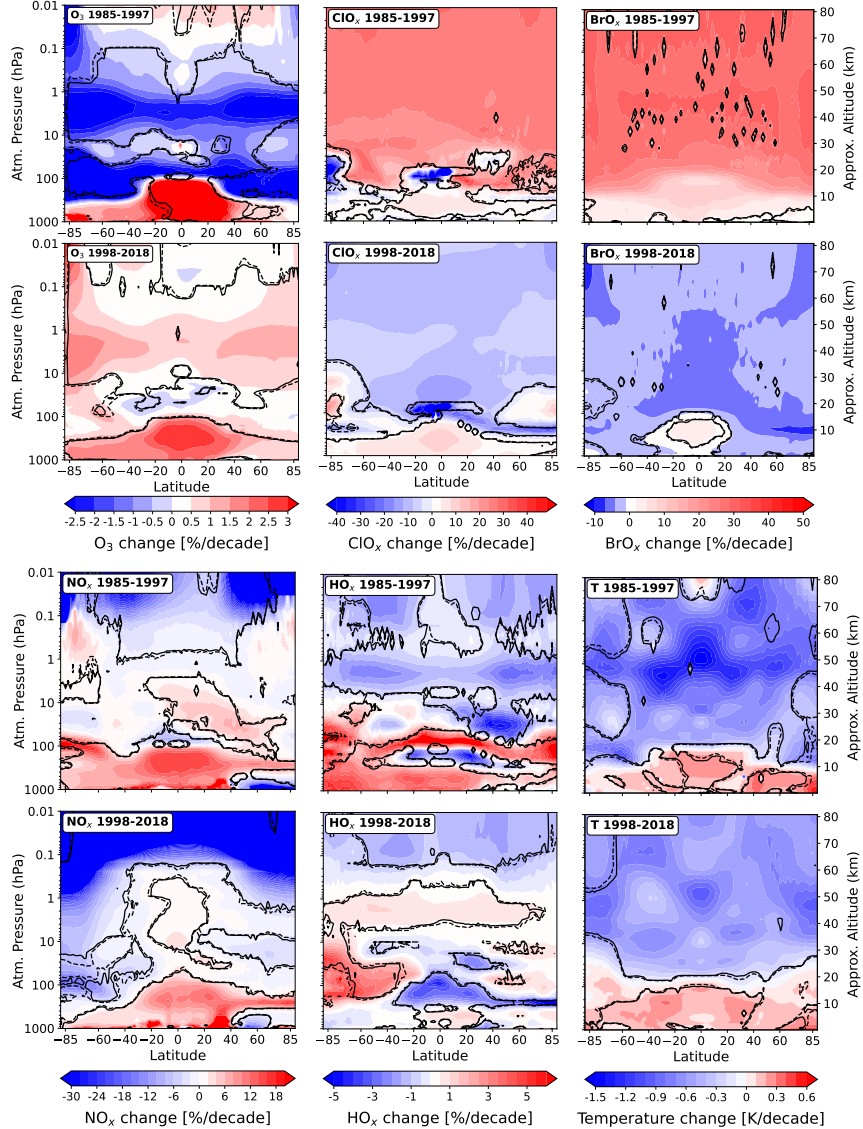

**Figure 2.** Ensemble mean trends of $O_3$, $ClO_x$, $BrO_x$, $NO_x$, $HO_x$ (%/decade) and temperature (K/decade) from the SOCOLv4 results. The dashed line is the delimiter of the region with significance at the 90% level for positive or negative changes; the solid line is the same at the 95% level.

The radicals shown here are odd nitrogen $[NO_x] = [NO] + [NO_2]$, odd hydrogen $[HO_x] = [OH] + [HO_2]$, reactive chlorine $[ClO_x] = [Cl] + [ClO]$, reactive bromine $[BrO_x] = [Br] + [BrO]$. In the troposphere, an upward trend in $NO_x$ is observed for






both periods, and it is stronger in the tropics. This positive trend is primarily due to the continuous increase in lightning activity and intensification of thunderstorms caused by global warming (Shindell et al., 2006) because the increase of the lighting flashes frequency produces additional $NO_x$ (Chameides et al., 1977; Sauvage et al., 2007). The $NO_x$ due to aircraft emissions may also contribute to the positive trend in upper tropospheric $NO_x$ (Köhler et al., 1997). Also, some $NO_x$ decrease related

to the improved air quality is visible over the Northern Hemisphere. In addition, more intensive convective mixing can also intensify the upward transport of ozone precursors. In the mesosphere, $NO_x$ strongly decreases in both periods. This is not related to solar activity, as its contribution is excluded by DLM. It can be explained by the suppressed $NO_x$ production due to the $CO_2$-related cooling in the mesosphere, which accelerates cannibalistic $NO + N \rightarrow N_2 + O$ reaction. The same can be said about the change in $HO_x$ in the mesosphere, which also has a negative trend, which is more pronounced and statistically

significant in the region where the temperature decreases. In the upper stratosphere, $HO_x$ trends follow ozone behavior because the $HO_x$ production from water vapor strongly depends on the $O(^1D)$ produced from the ozone. In addition, the decline in $HO_x$ is observed in the upper tropical troposphere, which may be related to some dehydration processes (Ueyama et al., 2014). The simulated increase in $ClO_x$ and $BrO_x$ in the free tropical troposphere may be associated with an increase in emissions of very short-lived substances from the ocean and intensification of upwelling. In the middle atmosphere, both $ClO_x$ and $BrO_x$

show a statistically significant increase during the ozone depletion period and a decrease during the ozone recovery period. The temperature decreases in the middle atmosphere, with a larger decline in the ozone depletion period. In the troposphere, the temperature rises everywhere, with a stronger increase in the tropics and in the Northern Hemisphere.

## 4.2 Partial and total column ozone evolutions for the 1985-2018 period

Figure 3 displays a comparison of the near-global [55°N-55°S], monthly mean anomalies in partial and total column ozone

for the 1985-2018 period simulated with SOCOLv4 and obtained from BASIC ozone composite as well as MERRA-2, ERA-5, and MSRv2 reanalysis. The reanalysis and ozone composite data are shown for atmospheric layers, for which they are available. Near-global tropospheric ozone shows a pronounced increase throughout the whole period with ∼4DU/period, which is facilitated by an increase in the number of ozone precursors, mainly $NO_x$ (see Figure 2). It is interesting to note that the mesospheric ozone has a tiny positive contribution to the recovery of the total column ozone on the near-global scale, as it

shows a positive trend of ∼0.03 DU for the 1998-2018 interval. The upper stratospheric ozone from both SOCOLv4 and BASIC demonstrates a pronounced decline, with a minimum after the Pinatubo eruption due to additional chlorine activation on volcanic aerosols. After 1998, the upper stratospheric ozone from both SOCOLv4 and BASIC started to recover distinctly. In the middle stratosphere, SOCOLv4 shows a pronounced increase in ozone for several years after the Pinatubo eruption due to an additional heterogeneous reactive uptake of $N_2O_5$ onto sulfuric acid particles, leading to a suppression of the $NO_x$

catalytic ozone cycle (Prather, 1992; Solomon, 1999; Rozanov et al., 2002). The reason why BASIC does not show a similar increase in ozone might be because of reported problems in satellite ozone retrieval during strong stratospheric aerosol loading (Davis et al., 2016). Unlike SOCOLv4, MERRA-2 and ERA-5 show a decrease in ozone over several years after the Pinatubo eruption. Also, in ERA-5, the start of ozone recovery is bias-shifted, showing the recovery only from the beginning of the 2000s. In the lower stratosphere, the ozone decline is seen until the mid-90s in all datasets. The ozone minimum in this region



occurred in 1992 because of the enhancement of chlorine activation after the Pinatubo eruption (Solomon et al., 1993). It should be mentioned that the Pinatubo affects ozone stronger in SOCOLv4, than seen in BASIC, which also might be due to biases in observations or a broader ozone hole area, which is seen in SOCOLv4 even over mid-latitudes (Sukhodolov et al., 2021). However, MERRA-2 and ERA-5 reanalysis show even stronger decreases caused by Pinatubo effects. Starting from 1996, the LSO evolution is rather uncertain as ozone there is highly perturbed. The DLM fits of LSO evolution from

SOCOLv4 are wavering around zero. BASIC shows the continuous decline of near-global LSO throughout the whole period, which is estimated to be 2DU since 1998. In ERA-5 and MERRA-2, ozone evolution has completely opposite behavior during the last years of the period, showing either decline or increase, correspondingly. ERA-5 and SOCOLv4 generally match each other for all presented partial ozone columns, and only a few years toward the end of the period ERA-5 shows a stronger ozone increase than SOCOLv4. MERRA-2 in the troposphere is not shown because of essential artifacts from unknown origins.

For higher atmospheric levels, MERRA-2 shows better reliability, despite the moderate agreement with the model and other datasets. ERA-5 is more applicable, but also shows biases, especially towards the end of the considered period. In general, the estimated DLM fits of total column ozone evolution modeled with SOCOLv4 are well within the range of fits from other data used. SOCOLv4 total column ozone evolution agrees better with MSRv2. For ERA-5, we see a much stronger increase in total column ozone than in the other data presented. MERRA-2 shows the marginal decline of total column ozone compared to

SOCOLv4 and other re-analysis, which may also be due to bias. Nevertheless, the presented time sequences display a recovery of the near-global total column ozone. Also, the important message is that the ERA-5 and MERRA-2 reanalyses still do not fit well for ozone trend analysis. Since they are a merging of observations and model data, there might be some biases related to model data or observations, preventing appropriate ozone trend analysis, which is also demonstrated by a large divergence of ozone trends between reanalysis data.



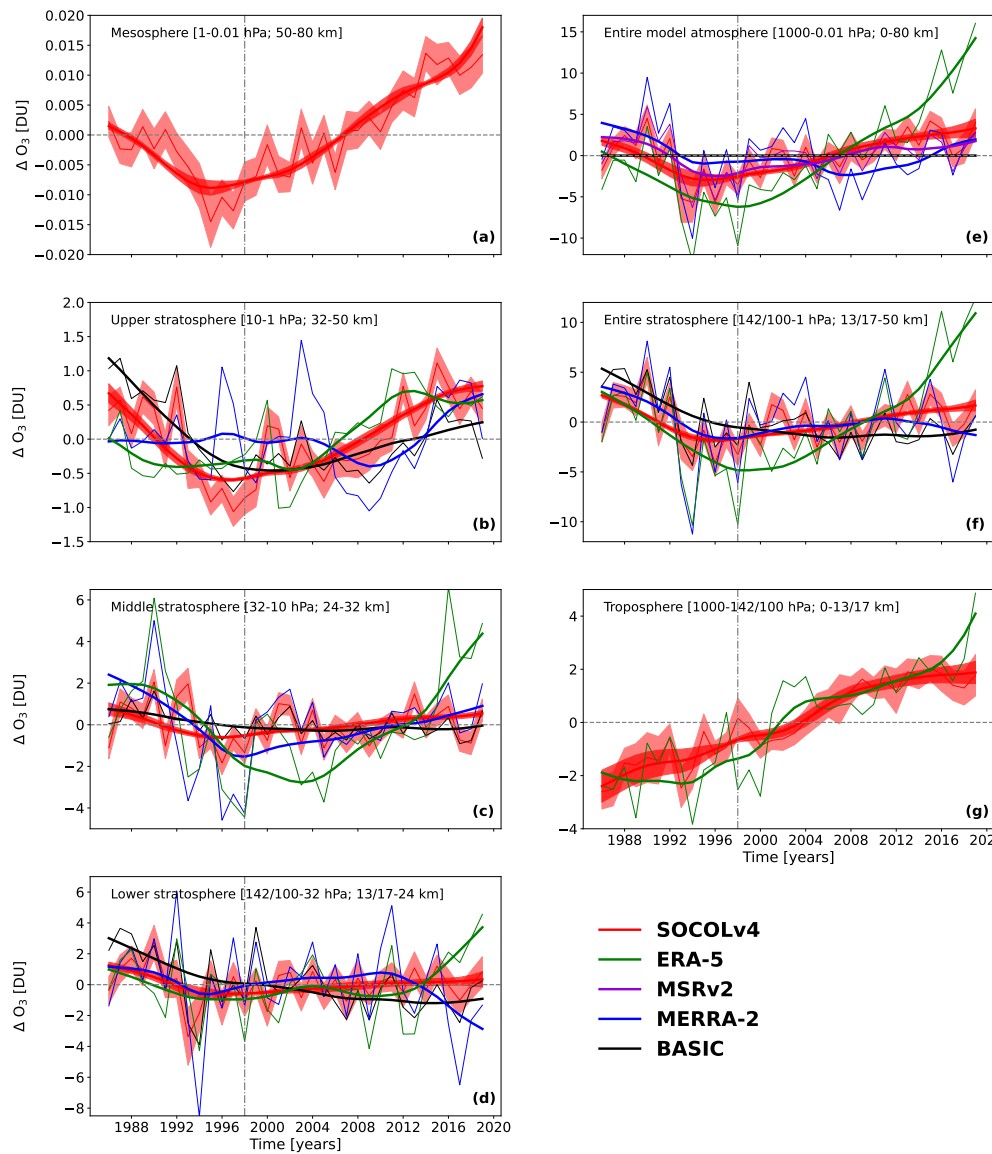

**Figure 3.** Near-global [55°N-55°S] annual-mean evolution of partial column ozone changes (in Dobson Units, DU) and DLM fits for the 1985-2018 period simulated with SOCOLv4 (red) and from the Bayesian ozone composite BASIC (black), from the reanalyses MERRA-2 (blue), ERA-5 (green), and MSRv2 (purple) for different atmospheric levels. Column ozone evolution and DLM fit presented for the **(a)** mesosphere; **(b)** upper stratosphere; **(c)** middle stratosphere; **(d)** lower stratosphere; **(e)** entire model atmosphere; **(f)** entire stratosphere, and **(g)** troposphere. The solid thin lines represent the ozone column anomalies; the solid curves represent the regression model fits computed by DLM, marked with the same color as the ozone anomalies. Red shadings represent 2-σ standard deviation of ozone evolutions and DLM fits between ensemble members of SOCOLv4 results. Dash-dotted vertical gray line marks the year 1998; dashed horizontal gray line marks the zero level.





### 255 4.3 Simulated and observed long-term ozone trends for the decline and recovery periods

In this study, we applied the statistical model for the whole period [1985-2018] and afterwards ozone changes for two phases of ozone evolution were computed from the DLM results. Figure 4 illustrates zonal mean decadal ozone trends for the 1985-1997 interval in the ensemble mean SOCOLv4 reference experiment and from BASIC observational composite.

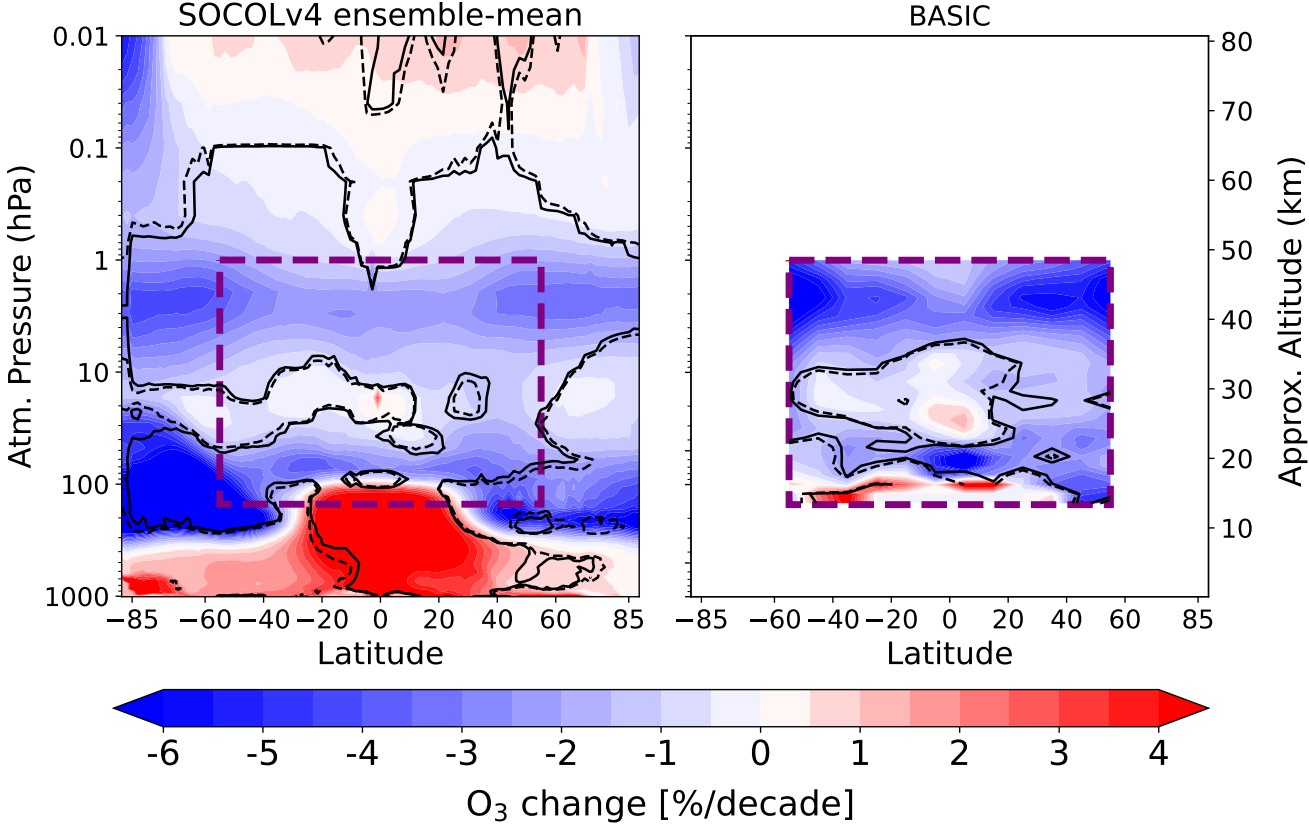

**Figure 4.** Ozone changes per decade (in %) for the ozone depletion period [between 1985 and 1997] from the ensemble mean SOCOLv4 reference experiment and BASIC observational composite. The dashed line is the delimiter of the region with significance at the 90% level for positive or negative changes; the solid line is the same at the 95% level. The purple dashed line indicates the region for which the BASIC composite is available.

The trend distribution calculated with SOCOLv4 is presented for the whole model atmosphere, i.e., from the ground to 0.01
hPa and for 90°N-90°S latitudes. For BASIC, the available region is only the near-global stratosphere [55°N-55°S]. In the lower atmosphere, below the tropopause, it is seen that the model tropospheric ozone has a strong positive and statistically robust trend. The positive trend might result from continuous emissions of tropospheric ozone precursors (Griffiths et al., 2021). The chemical reactions that involve the precursors lead to ozone formation. However, we would like to emphasize that the positive





trend of tropospheric ozone is getting stronger in the tropics [20°N-20°S], showing an increase of more than 4%/decade that

is attributed to increasing $NO_x$ (see Figure 2). In high latitudes of both hemispheres, strong negative changes in ozone are observed. Even if the Antarctic ozone hole is not a continuous event and occurs only during austral springtime, the pronounced and statistically significant negative changes of more than 6%/decade are seen in the Southern Hemisphere, indicating the effect of ozone hole intensification. The decline in ozone in the lower stratosphere of the Southern Hemisphere appears in SOCOLv4 even over the mid-latitudes [55°S-40°S]. As shown in (Sukhodolov et al., 2021), the expansion of the ozone hole in SOCOLv4

is larger than in observations. Most likely, it is related to atmospheric dynamics and transport. In the Northern Hemisphere, the negative lower stratospheric ozone changes also appeared. It should be noted that the strong ozone hole events over the Arctic are rather irregular, and usually, the Arctic ozone depletion is weaker than the one over the Antarctic (WMO, 2018). We obtained prominent, statistically significant negative ozone changes of about 4-5%/decade over mid- and high latitudes in the upper stratosphere of both hemispheres. This effect is observed because during this period the continuous increase in

halogens content strongly influenced the loss of ozone in these regions, and the catalytic ozone destruction cycle involving chlorine is more effective in the upper stratosphere due to the presence of a sufficient number of oxygen atoms (Chipperfield et al., 2018). Negative ozone trends from BASIC seen in the upper stratosphere and over tropical lower stratosphere exceed those from SOCOLv4, showing depletion in ozone even more than 6%/decade. As it was mentioned above, the notable feature can be seen in the mesosphere, where SOCOLv4 demonstrates positive changes in ozone. It might be resulting from the $HO_x$

decline (see Figure 2). The profile of decadal ozone trends for the 1998-2018 interval from all individual ensemble members and ensemble mean of SOCOLv4 reference experiment and BASIC observational composite are depicted in Figure 5.





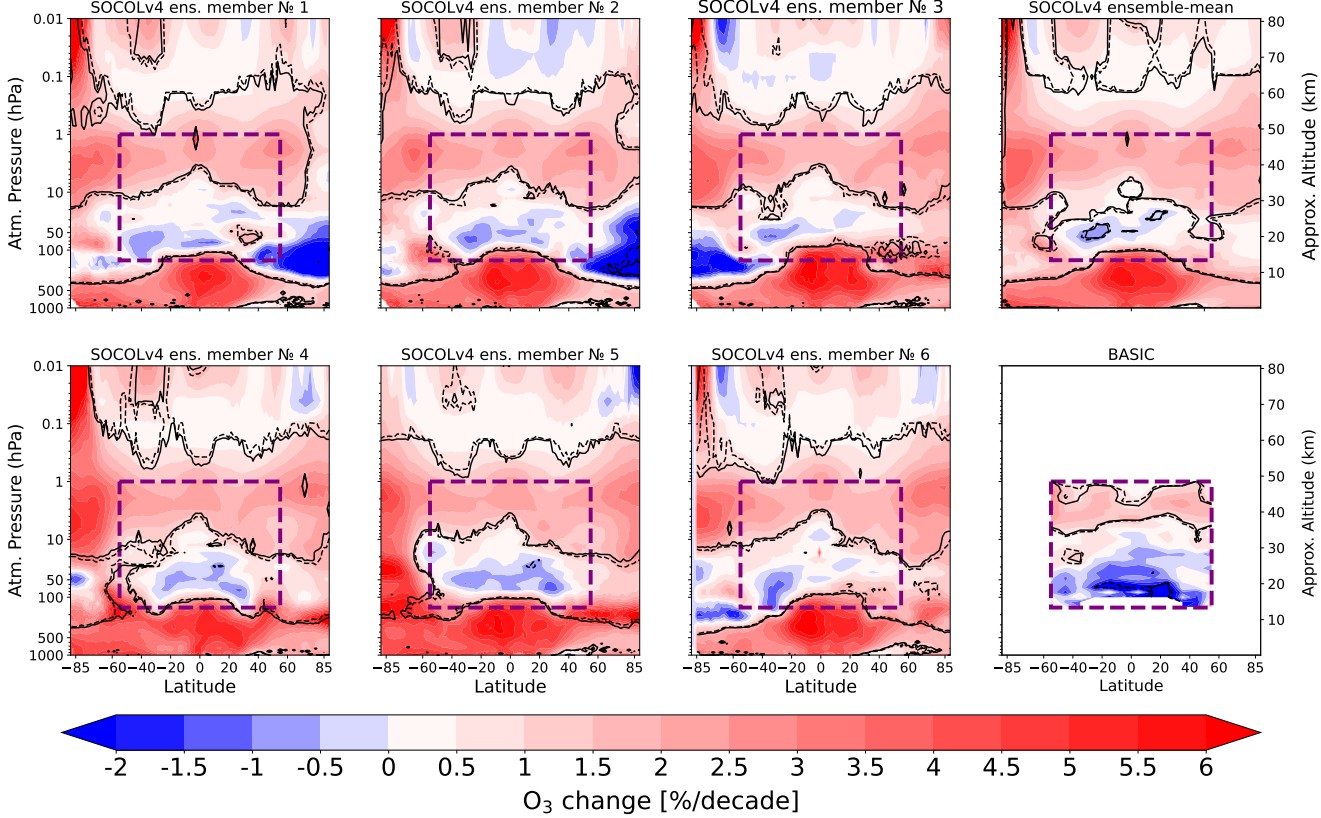

**Figure 5.** Zonal mean ozone changes for the ozone recovery period [between 1998 and 2018] from the SOCOLv4 ensemble experiment and the BASIC observational composite (in percent per decade). The name of each ensemble member (or the ensemble mean) is indicated at the top of each panel. The dashed line indicates the region with significance at the 90% level for positive or negative changes; the solid line is the same at the 95% level. The purple dashed line indicates the region for which the BASIC composite is available.

For the recovery period, the tropospheric positive ozone changes with a similar magnitude remained stable, as the production of ozone precursors continued to increase for this period too (see Figure 2). The tropical free-tropospheric ozone shows an increase of ∼4-5%/decade. The mesospheric ozone shows an upward trend with a higher significance over the mid-latitudes

of the Southern Hemisphere in most realizations, where a significant decline in $NO_x$ and $HO_x$ is observed (see Figure 2). We would like to emphasize again that the possible ozone recovery in the mesosphere might be important, since it can contribute to the total column ozone recovery and compensate for the ozone decline in lower atmospheric levels to some extent. In the upper stratosphere, the recovery of about 3-4%/decade over mid- and high latitudes is visible, statistically robust, and observed in all presented realizations of the SOCOLv4 reference experiment and BASIC composite as expected because of hODS limitation

by MPA, mainly $ClO_x$ (see Figure 2). The simulated ozone changes in the upper stratosphere are slightly overestimating the BASIC by ∼1% and have a more pronounced tendency to increase in a poleward direction. The middle stratospheric ozone





from both SOCOLv4 and BASIC demonstrates a near-neutral state, showing some signs of positive and negative changes without a high level of statistical significance. In the lower stratosphere, the ozone in high latitudes of both hemispheres is not showing the recovery in all presented ensemble members of the experiment and even demonstrating some negative changes.

In three ensemble members, there are regions with a manifestation of negative signal in ozone either in the northern or in the southern high latitudes, which might be artifacts as they are non-significant. The ensemble mean shows a slightly positive trend in LSO over high latitudes, but it is also non-significant. This may be related to an underestimation of ozone production or some dynamical biases in these ensemble members during the recovery period. The modeled extratropical LSO changes vary significantly between ensemble members. Some slightly negative trends of ~1-1.5%/decade are observed. However, the

statistical significance is less than 90% for a major part of the lower stratosphere. However, the ensemble-mean shows some statistically significant LSO trends. Yet, it should be stated that positive statistically significant trends in the extratropical lower stratosphere are also not observed and the ozone trends in this region are uncertain. The extratropical region of the lower stratosphere is a highly dynamic environment, and we see that ozone trends can vary much from one ensemble member to another.

## 5 Discussion and summary

In this paper, we examine long-term ozone trends computed with the Earth System Model (ESM) SOCOLv4 in an ensemble with six members of a reference experiment and compare these realizations to data from the BASIC composite and from various reanalyses. The evolution of long-term ozone changes is quantified using dynamical linear modeling. In general, the ozone trends integrated across atmospheric layers (i.e., troposphere; lower, middle and upper stratosphere; mesosphere) are well

captured by SOCOLv4, although there are also some discernible deviations from the available observations and reanalyses. The model shows a continuous ozone increase during the entire period in the troposphere and for the recovery phase in the mesosphere, which is important to consider, because both contribute positively to the overall recovery of the total column ozone. The modeled total column ozone shows good agreement with the reanalyses. We also show that the MERRA-2 and ERA-5 reanalyses are less suitable for ozone trend analysis because they have artifacts from unknown origin that interfere with

a correct trend estimate, possibly because the reanalyses are affected by simulations made with simplified chemistry models or because they apply not yet homogenized satellite observations. Although ozone trends in SOCOLv4 are well simulated in most atmospheric layers, our study infers that the modeled ozone changes in the extratropical lower stratosphere are still uncertain. While the model provides regions with negative trends in LSO also in the ozone recovery period [1998-2018], the regions are patchy, and the trends are weak and non-significant, i.e., do not agree well with the observed ozone changes obtained

from the BASIC. The level of uncertainty is high, as evidenced by LSO trends varying a lot between individual ensemble members. However, no single member (amongst the six) suggests an extensive negative signal found in the BASIC. Indeed, detecting clear and robust ozone trends outside the polar regions in model simulations is difficult because of the high dynamical variability perturbing the individual ensemble members (Stone et al., 2018). This justifies analyzing each ensemble member separately to study extratropical LSO variability, as we do in the present work. Two of the ensemble members (No. 4 and 5)



show resemblance with BASIC in the lower stratosphere during the recovery period, including extensive regions with negative ozone trends. Yet, the other four members have less pronounced but still negative trend regions. And for all the six members, the strength of the modeled negative trends remains much lower than seen in BASIC. Consequently, the SOCOLv4 ensemble-mean shows only weak but statistically significant trends. Recently, Godin-Beekmann et al. (2022) provided updated trends of the stratospheric ozone vertical distribution based on the LOTUS regression model, which relies on merged satellite records tested against ground-based records (but not using DLM). They confirmed the large influence by short-term dynamical variability, preventing solid conclusions on ozone trends in the extratropical lower stratosphere. Even small changes in time series duration ($\pm$ 1 year) can affect the trend patterns (Ball et al., 2019). However, adding three more years between the original version of LOTUS and the update still did not produce statistically significant trends in LSO for the recovery phase. Possibly, the ozone recovery period is not yet long enough to detect statistically significant LSO trends in the LOTUS regression analysis (Godin-Beekmann et al., 2022). The attribution of all necessary regressors is also important to increase the robustness of the trend. However, also the BASIC observational composite might have some limitations based on the quality and accuracy of measurements included in it. Another very recent paper by Bognar et al. (2022) describes a new merged ozone profile data set, based on SAGE II, SAGE III / ISS, and updated OSIRIS data. Trends are determined by using and comparing MLR and DLM analyses. For the 2000–2021 period, they show that ozone decreased by 1–3% in the lower stratosphere since 2000. These decreases were found to be more pronounced in DLM than in MLR, and significant in the tropics (>95% confidence), but not necessarily at mid-latitudes (>80% confidence). They further highlight that in tropopause relative coordinates, most of the negative trends in the tropics lose significance, highlighting the impacts of a warming tropopause and increasing tropopause altitudes. Similar to the present work, their findings confirm the poorly understood continuing decline of ozone in parts of the extratropical lower stratosphere. We should emphasize the need to find out what the ozone changes in the extratropical lower stratosphere are related to. The main influences may be related to uncertainties in emissions, in particular on greenhouse gases (GHG), ozone-depleting substances (ODS), anthropogenic very short-lived substances (VSLS) (Alejandro Barrera et al., 2020) including iodine containing species (Karagodin-Doyennel et al., 2021), and volcanic aerosol loading. Other factors might concern structural uncertainties of the models, such as their horizontal and vertical resolution in the upper troposphere and lower stratosphere. In summary, the trends identified by SOCOLv4 are largely consistent with previous findings and confirm the general understanding of ozone recovery, including the effects of climate change on the ozone layer. The results further confirm the poorly understood ongoing decline of ozone in parts of the extratropical lower stratosphere and show that also an ensemble approach, while capturing the high dynamical variability in this region, does not yield the observed negative trends. The presented results indicate that the obtained qualitative agreement between the model and observations allows to produce reliable estimates of future ozone evolution using modern chemistry-climate models.

*Acknowledgements.* A.K.-D., E.R., T.S., J.S., and T.E. thank the Swiss National Science Foundation for supporting this research through the №200020-182239 project POLE (Polar Ozone Layer Evolution). Calculations were supported by a grant from the Swiss National Super-computing Centre (CSCS) under projects S-901 (ID 154), S-1029 (ID 249), and S-903. Part of the model development was performed on



the ETH Zürich cluster EULER. We are also grateful to M. Laine for the opportunity to use the dynamical linear regression model in our study. The work of Eugene Rozanov and Timofei Sukhodolov was performed in the SPbSU "Ozone Layer and Upper Atmosphere Research"

laboratory, supported by the Ministry of Science and Higher Education of the Russian Federation under agreement 075-15-2021-583.

*Data availability.*   The SOCOLv4 model data can be downloaded from https://doi.org/10.5281/zenodo.6885544   (Karagodin-Doyennel , 2022). The homogenized BAyeSian Integrated and Consolidated (BASIC) composite ozone time series are freely available at https://data.mendeley.com/datasets/2mgx2xzzpk/3   (Ball et al., 2017; Alsing and Ball, 2019). The Multi-Sensor Re-analysis data version 2 (MSRv2) reanalysis dataset can be downloaded from https://www.temis.nl/protocols/O3global.php  (van der A et al., 2015). The Modern-Era

Retrospective analysis for Research and Applications, Version 2 (MERRA-2) reanalysis dataset is available at https://disc.gsfc.nasa.gov/datasets?project=MERRA-2   (Gelaro et al., 2017). The 4D-Var data assimilation-based comprehensive European ReAnalysis (ERA-5) reanalysis dataset can be found here: https://cds.climate.copernicus.eu/!/search?text=ERA5type=dataset   (Hersbach et al., 2020).

*Author contributions.*   AKD prepared the original draft, conducted the entire DLM analysis, and visualized the results. ER and TP supervised

the whole study. WB and ER originated the idea for this study. TS, JS, and AKD designed and run the SOCOLv4 reference experiment. All authors participated in manuscript editing and discussions about the results.

*Competing interests.*   The authors declare that they have no conflict of interest.



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





## Appendix A



**Figure A1.** Input quantities (proxy variables) for the forcing of the SOCOLv4 ensemble experiment.





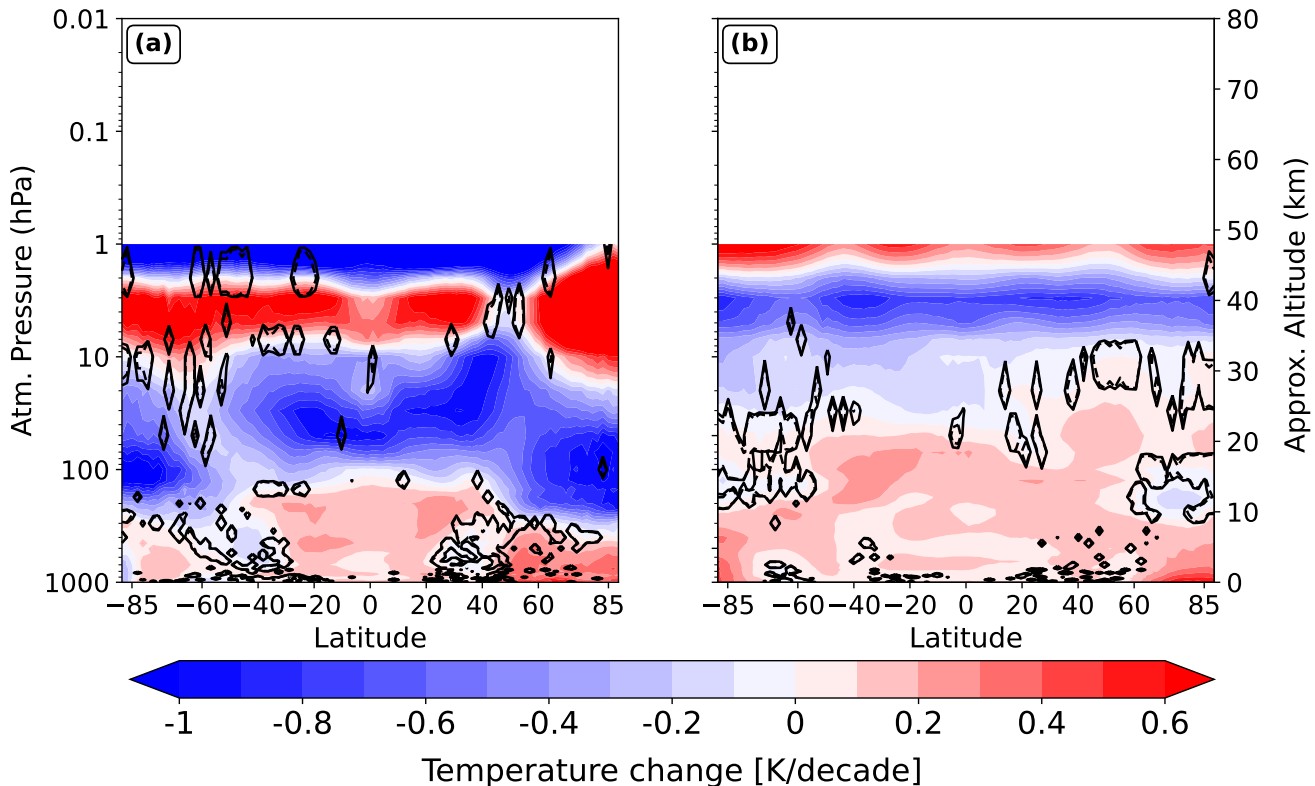

**Figure A2.** Profiles of temperature trends (K/decade) obtained from the ERA-5 reanalysis dataset. **(a)**: ozone depletion period [1985-1997]; **(b)**: ozone recovery period [1998-2018]. The dashed line indicates the region with significance at the 90% level for positive or negative changes, the solid line is the same at the 95% level.