# Peer review of "The historical ozone trends simulated with the SOCOLv4 and their comparison with observations and reanalysis"

_EGUsphere, 2022_

## Referee Comment (RC1)

Review of the Paper :  "The historical ozone trends simulated with the SOCOLv4 and their comparison with observations and reanalysis" (egusphere-2022-701),

by Arseniy Karagodin-Doyennel, Eugene Rozanov, Timofei Sukhodolov, Tatiana Egorova, Jan Sedlacek, William Ball+, and Thomas Peter

General comments:

The paper addresses the very interesting topic of stratospheric ozone trends and variability after 1985, split in two periods (i.e., ozone decline and recovery), with a focus on the lower stratosphere where a continuing decline of ozone in the lower tropical and mid-latitude atmosphere has been the subject of recent works. The data used in the paper are derived from Chemistry-Climate Model simulations, and a comparison of trends derived from observations composites and reanalysis data from different sources is performed.

The subject is appropriate to the Journal and contains significant original material.

The paper is generally well organized, but there are parts that need to be revisited for sake of clarity of meaning.

I recommend publication after minor revisions.

Specific comments below:

1.   **Section 3 (description of the DLM approach)**

This section, especially in the last paragraph (line 182 and below should be revisited to clarify the meaning. It would be better if you split this paragraph in two parts (from sentence beginning with "It is important to note…") so that you may explain better how the trends were calculated ("..calculated from the DLM output. ..").

Was the same approach followed for the reanalysis datasets? How was it done?

Please write a line also to comment on the assessment of significance.

In the same paragraph, lines 181-182, please explain the "within 0.20-0.25". Are the correlation coefficients always positive for all variables? Or you mean something different?

2.   **Section 4. 2 (Partial and total column ozone evolutions for the 1985-2018 period)**

This section should be re-written for a clearer meaning allover.

You start in line 222 with tropospheric ozone, which is then dropped (and revisited later for the observations), and in the same line (223), immediately after the end of the sentence you note changes in mesospheric ozone.

Please reorganize the paragraphs, so that you facilitate reading and clarify your findings, especially the comparison to reanalysis. It is in this section you need to justify the statement in **Section 5** (line 313 -) "...We also show that the MERRA-2 and ERA-5 reanalyses are less suitable for ozone trend analysis because..."

The same comment for reorganization of paragraphs applies to all remaining sections, as it might even be confusing at some points.

3.    The statement in the last lines of **Section 5** (line 350 -)

"The results further confirm the poorly understood ongoing decline of ozone in parts of the extratropical lower stratosphere…"

What do the results confirm? That there is a continuous ozone decline in the extratropical lower stratosphere (statistically significant), the origins of which are poorly understood, or that there are indications of an ozone decline, with a patchy response, with origins that are not understood?

So please rephrase to clearly present your findings and their importance.

**Technical and other comments**

**Abstract**

Line 9: "…derived from observations and reanalyses." Please refer here to the datasets, e.e. "namely the BASIC composite of ozone, and…"

Line 14 "…do not agree with some observation composite analysis." Which ones? Specify

**Introduction**

Line 30 "… no or a …" change to "none or a …"

Line 69 "…BDC…" please give the name in full before using acronym for the first time, the reference to Butchart et al could be given here as well.

**Section 3.  The description of the DLM approach**

Line 168 "…are used to represent trends…" what is the meaning of these last words?  Please clarify.

Line 184 "…output DLM output." Please delete the first word

**Section 4.2**

For clarity, in line 219 please write "and total column ozone (represented in Figure 3 by the entire model atmosphere, upper right panel)…" or something similar.

Line 236-237 "… might be due to …" --> "… might be due to either…or…"

**Section 4.3.**

Line 256 "In this study we applied … and afterwards ozone changes..." please change to "In this study, we first applied… and then computed ozone changes…"

Line 261 "However…" Why "however"??

The same for lines 299 -300, you use "However…" twice

**Section 5**

Line 308 "…various reanalyses." -> "… a number of available reanalyses."

Line 318 "…the model provides…" -> "the model shows"?

Line 351 "…and show that also an ensemble approach…"  could be better as "… moreover show that even an ensemble approach…"

---

## Author Comment (AC1)

**Referee #1**

*Dear Referee,*

*We are grateful for reviewing our manuscript and your valuable comments and suggestions, which will largely improve our manuscript. Below you can find responses to each of your comments (marked in blue):*

General comments:

The paper addresses the very interesting topic of stratospheric ozone trends and variability after 1985, split in two periods (i.e., ozone decline and recovery), with a focus on the lower stratosphere where a continuing decline of ozone in the lower tropical and mid-latitude atmosphere has been the subject of recent works. The data used in the paper are derived from Chemistry-Climate Model simulations, and a comparison of trends derived from observations composites and reanalysis data from different sources is performed.

The subject is appropriate to the Journal and contains significant original material.

The paper is generally well organized, but there are parts that need to be revisited for sake of clarity of meaning.

I recommend publication after minor revisions.

Specific comments below:

1. **Section 3 (description of the DLM approach)**

This section, especially in the last paragraph (line 182 and below should be revisited to clarify the meaning. It would be better if you split this paragraph in two parts (from sentence beginning with "It is important to note…") so that you may explain better how the trends were calculated ("..calculated from the DLM output. ..").

Thank you for your comment. This part has been largely rewritten for clarity and understandability as follows:

"The remaining proxies show the average correlation coefficients between -0.2 and 0.2, which is considered to be a "weak" mutual dependency.

It is important to note that in this study, all DLM calculations using model data as well as ozone composite and reanalyzes were performed for the entire 1985-2018 period. The evolution of the predicted variable is characterized in DLM by the so-called "trend-term" or background level. In simple terms, it is the evolution of a variable filtered from the known part of its variability, induced by external forcings, which are represented in DLM by proxies. The state of the predicted variable for the one-step-ahead is predicted by the Kalman filter, and the Kalman smoother is used for the marginal probability distribution of the state. The Markov Chain Monte Carlo (MCMC) method (Alsing, 2019) is used to infer the posterior distributions of the background level. Then, 200 samples of the background level were

drawn from the DLM states, which describe the posterior distribution uncertainty. It was done to determine the standard deviation of background level between these samples used to calculate the statistical significance of the results. Afterwards, the ozone trends per decade are calculated separately for the phases of the ozone evolution ([between 1985 and 1997] and [between 1998 and 2018]) from the mean background level by applying the conventional linear regression ($\alpha x + \beta$). In linear regression, $\alpha$ means a slope term, and, hence, a trend per decade at each grid point (latitudes x heights) are calculated as: $\alpha$*length of month-to-month time series/number of decades. Finally, the statistical significance is determined for each ensemble member using the mean background level and its standard deviation between MCMC samples by the Student's t-test. The ensemble mean trends of the SOCOLv4 experiment are calculated as an average of the trends from all individual ensemble members. The statistical significance for the ensemble mean trends is calculated using a standard deviation of trends between individual ensemble members. Trends in BASIC ozone composite and reanalysis data sets were calculated using the same methodology but applying observed proxy variables in DLM, the same as presented in Ball et al. (2018) as it was mentioned above. "

Was the same approach followed for the reanalysis datasets? How was it done?

Yes, the methodology is the same. However, to analyze the ozone from BASIC observational composite and all reanalyzes, DLM uses observed proxy variables, as in Ball et al. (2018).

Please write a line also to comment on the assessment of significance.

The statistical significance of the trends was calculated using the Student's t-test. It was done for all individual ensemble members of the SOCOLv4 reference experiment as well as BASIC observational composite and reanalyses using the sample mean background level and its standard deviation between MCMC samples. In the case of the ensemble mean of the SOCOLv4 reference experiment, statistical significance was calculated using the ensemble mean trend and its standard deviation between ensemble members of the experiment.

In text we included this as follows:

"Finally, the statistical significance is determined for each ensemble member using the mean background level and its standard deviation between MCMC samples by the Student's test. The ensemble mean trends of the SOCOLv4 experiment are calculated as an average of the trends from all individual ensemble members. The statistical significance for the ensemble mean trends is calculated using a standard deviation of trends between individual ensemble members."

In the same paragraph, lines 181-182, please explain the "within 0.20-0.25". Are the correlation coefficients always positive for all variables? Or you mean something different?

Thank you for the comment, it is, indeed, not so clear what is meant by "within 0.20-0.25" here. In fact, Figure 1 shows the correlation coefficients between proxies for model data.

They are generally between -0.2 and 0.2. In the text, this line has now been rephrased to be clearer.

2. **Section 4. 2 (Partial and total column ozone evolutions for the 1985-2018 period)**

This section should be re-written for a clearer meaning allover.

You start in line 222 with tropospheric ozone, which is then dropped (and revisited later for the observations), and in the same line (223), immediately after the end of the sentence you note changes in mesospheric ozone.

Thank you for your comment. This paragraph has been fully rewritten and now evolutions of ozone from all used data sets are described layer-by-layer so that readers can easier follow the text. The revised paragraph is below:

[revised manuscript text omitted]

Please reorganize the paragraphs, so that you facilitate reading and clarify your findings, especially the comparison to reanalysis. It is in this section you need to justify the statement in **Section 5** (line 313 -) "...We also show that the MERRA-2 and ERA-5 reanalyses are less suitable for ozone trend analysis because..."

Thank you for this comment. Indeed, in Figure 3 it is seen that ozone evolutions from both MERRA-2 and ERA-5 reanalyzes are generally biased in the stratosphere against BASIC observational composite and SOCOLv4. We cannot demonstrate the source of this discrepancy and it is out of the scope of this paper, but it might stem from some issues with their underlying models, specific observational data processing, or assimilation approaches. This has now been properly addressed in the rewritten paragraph (see the rewritten paragraph in just above comment).

The same comment for reorganization of paragraphs applies to all remaining sections, as it might even be confusing at some points.

Agreed. The next sections have also been restructured to have the description layer-by-layer (see reformulated subsection 4.3.  Simulated and observed long-term ozone trends for the decline and recovery phases and section 5 Discussion and summary.)

3.  The statement in the last lines of **Section 5** (line 350 -)

"The results further confirm the poorly understood ongoing decline of ozone in parts of the extratropical lower stratosphere…"

What do the results confirm? That there is a continuous ozone decline in the extratropical lower stratosphere (statistically significant), the origins of which are poorly understood, or that there are indications of an ozone decline, with a patchy response, with origins that are not understood?

So please rephrase to clearly present your findings and their importance.

Rather the second.  We rephrased this part as follows: "The results further confirm that there are marginally significant negative ozone changes in parts of the low latitude lower stratosphere. This result agrees in general with the negative trends extracted from satellite data composite, however the simulated magnitude and significance are lower than in observations."

**Technical and other comments**

**Abstract**

Line 9: "…derived from observations and reanalyses." Please refer here to the datasets, e.e. "namely the BASIC composite of ozone, and…"

Agreed. It has been reformulated as follows:

"We applied the new Earth system model SOCOLv4 to calculate long-term ozone trends and compare them with trends derived from BASIC ozone composite and MERRA-2, ERA-5, and MSRv2 reanalyzes."

Line 14 "…do not agree with some observation composite analysis." Which ones? Specify

This sentence has been modified: "….with BASIC ozone composite."

**Introduction**

Line 30 "… no or a …" change to "none or a …"

Done.

Line 69 "…BDC…" please give the name in full before using acronym for the first time, the reference to Butchart et al could be given here as well.

Agreed. It has been rewritten like this: "…the meridional Brewer-Dobson circulation (BDC, Butchart et al., 2006)….".

**Section 3.  The description of the DLM approach**

Line 168 "…are used to represent trends…" what is the meaning of these last words?  Please clarify.

We agree that it was unclearly formulated. We amended it like this: "…are used to explain the variability in total/partial ozone."

Line 184 "…output DLM output." Please delete the first word

Done.

**Section 4.2**

For clarity, in line 219 please write "and total column ozone (represented in Figure 3 by the entire model atmosphere, upper right panel)…" or something similar.

Done.

Line 236-237 "… might be due to …" --> "… might be due to either…or…"

Done.

**Section 4.3.**

Line 256 "In this study we applied … and afterwards ozone changes..." please change to "In this study, we first applied… and then computed ozone changes…"

Done.

Line 261 "However…" Why "however"??

Agree, it is unneeded here and has been deleted from the text.

The same for lines 299 -300, you use "However…" twice

Yes, the one of them is deleted.

**Section 5**

Line 308 "…various reanalyses." -> "… a number of available reanalyses."

Done.

Line 318 "…the model provides…" -> "the model shows"?

Done.

Line 351 "…and show that also an ensemble approach…"  could be better as "… moreover show that even an ensemble approach…"

Done.

---

## Author Comment (AC2)

**Referee #2**

Review of "The historical ozone trends simulated with the SOCOLv4 and their comparison with observations and reanalysis", by Karagodin-Doyennel

GENERAL COMMENTS

I would like to start this review by acknowledging the passing of William Ball, one of the co-authors on the paper, a wonderful person and a great scientist. He will be sorely missed within the ozone research community.

This is a relatively straightforward paper that makes use of the SOCOLv4 Earth System Model to simulate changes in ozone from 1985 to 2018 with a focus on understanding trends in lower stratospheric ozone and how they change between the "ozone depletion phase" (1985-1998) and the "ozone recovery phase" (1998-2018). I have made some suggestions for corrections below. Once these have been addressed, the paper will be suitable for publication in EGUsphere.

*Dear Referee,*

*We are grateful to the reviewer for his/her warm words and the favorable review as well as appreciate for a large amount of time he/she has spent preparing the detailed comments and suggestions to improve our manuscript. Our specific responses to each comment have been given in the text and below (marked in* blue*):*

SPECIFIC COMMENTS

Line 2: On the use of the word "recover" and "recovery": Whenever the word is used it needs to be made clear what ozone is recovering from. In this case, presumably, from the effects of ozone depleting substances (ODSs). Then a clear distinction needs to be made between (1) ozone increasing, and (2) ozone recovering from the effects of ODSs. One may occur without the other. Ozone in the tropical lower stratosphere may be decreasing while still recovering from the effects of ODSs. So, let's see how this term is used through the remainder of the paper. But here, in the first line of the abstract, I want you to be clear as to whether you are referring to (1) ozone increasing, or (2) ozone recovering from the effects of ODSs (which then requires a clear attribution to decreasing concentrations of ODSs).

Thank you for this remark. Indeed, the reader may not understand the meaning of the words "recovery" and "increase" without a proper context. As such, in this very line, we, of cause, meant the recovery from the effect of hODS. In the case of tropospheric ozone, the increase is due to the anthropogenic NOx and CH4. Both have now been amended in this line and in similar cases throughout the text to make it clearer.

Line 3: Replace "Amendments" with "Amendments and Adjustments". Likewise on line 20.

Done.

Line 5: So, in this regard, here is what I wrote in a review of a William Ball paper in 2019 as I suspect it is going to be relevant to the rest of this review:

"So are you really saying that the Montreal Protocol is working only in the upper stratosphere and not in the lower stratosphere? This will hugely concern policy-makers. They will wonder why all the hard work they have done since 1987 in reducing emissions of CFCs, halons, HCFCs and other ODSs has only decreased their concentrations in the upper stratosphere. Could I put it to you that the Montreal Protocol has been effective in reducing ODS concentrations, and thereby concentrations of Cly and Bry throughout the atmosphere, and that, as a result, ozone throughout the atmosphere, including the lower stratosphere, is recovering from the effects of those ODSs. Is this recovery apparent in observations in the upper stratosphere? Apparently yes. I say apparently only in that (at least in this paper) a thorough attribution of the drivers of those ozone increases has not been done. Is this recovery apparent in observations in the lower stratosphere? No, clearly not? Why not? Well because other factors have been affecting ozone (not diagnosed in this paper) that are likely (we cannot be sure since a thorough attribution has not been done) overwhelming the increases brought about by reductions in concentrations of Cly and Bry. Wouldn't that be a more accurate picture to communicate to policy-makers?"

Please note that that long comment was in response to the Ball 2019 paper and not to the current paper. Your statement that "continuing decline in the lower tropical and mid-latitudinal stratospheric ozone" is, in no way, an indictment of the MPA.

We apologize for the ambiguity. In essence, we did not imply any downplay of the undoubted role of MPA in saving the ozone layer. We just say that there is evidence of the declining trend in lower stratospheric ozone in the extra-polar region, which is not related to the MPA "bad work" but is very likely dynamically driven. And, as you also mentioned about "recovery" and "increase" conflation, here we would say that even in the lower stratosphere, thanks to MPA, the ozone recovers from the effects of hODS, despite the revealed signs of its decline, caused by other effects.

Line 10: Yes, I think it is valid to refer to 1998-2018 as the "ozone recovery" period since that is when stratospheric loading of Cly and Bry was decreasing and ozone was recovering from its effects (even if ozone was not everywhere increasing).

We agree that this is a valid naming for these intervals of ozone evolution during the long 1985-2018 period.

Line 11: Does SOCOLv4 show clear ozone recovery or does it show statistically significant positive trends in ozone? They are NOT the same thing since ozone could be decreasing but still recovering from the effects of ODSs. The first (ozone recovery) requires *attribution* to declining concentrations in Cly and Bry. The second (statistically significant positive trends) is purely the result of statistical analyses and requires no attribution - strictly a *detection* issue. Please always be clear throughout this paper which you are referring to and do not conflate the two.

Here, we also agree that this is unclear wording. Now, it has been reformulated properly as follows: "…SOCOLv4 shows statistically significant positive ozone trends in the mesosphere, upper and middle stratosphere …".

Lines 12-17: It feels like the last two sentences of the abstract contradict each other. The penultimate sentence says that SOCOLv4 and observations disagree. The last sentence says that there "is in general agreement with observations". This apparent inconsistency needs to be resolved.

"We reformulated this sentences as follows: "…Thus, the model results reveal marginally significant negative ozone changes in parts of the low latitude lower stratosphere, which agrees in general with the negative tendencies extracted from satellite data composite. Despite the slightly smaller significance and magnitude of the simulated ensemble mean, we confirm that modern chemistry-climate models such as SOCOLv4 are generally capable of simulating the observed ozone changes, justifying their use to project the future evolution of the ozone layer."

Line 20: I wouldn't say "are beginning to take effect". I would say "have clearly taken effect over the past 25 years".

Agreed. It has been amended.

Line 20: In this case I think that what you mean instead of "observable ozone recovery at certain latitudes and altitudes" is "observable ozone increases at certain latitudes and altitudes".

Yes, it should be "increase" here. "Recovery" has been exchanged with "increases".

Line 21: I think that it would be better to use the word "simulation" here rather than "projection". These studies weren't done to make projections in the same way that the IPCC models are used to make climate projections.

We agree that "simulation" would be better to use in this context. So, we replaced "projection" with "simulation" here.

Lines 23-24: I would suggest replacing "time of the ozone recovery back to the pre-1980 level for different regions" with "time of the return of ozone to pre-1980 levels for different regions".

Agreed.

Line 25: I would say mid to late 1990s depending on where you look in the stratosphere.

Yes, we agree, it is different for different layers. So, we rephrased it a bit as follows: "In the mid-to-late 1990s…"

Line 29: And now, again, you are conflating ozone recovery from the effects of ODSs with the return of ozone to pre-1980 levels. They are not the same thing. For example, tropical lower stratospheric ozone has been recovering from the effects of ODSs since the late 1990s, as you state. However, it is possible that tropical lower stratospheric ozone never returns to pre-1980 levels because other factors have been affecting ozone in that part of the atmosphere. Being clear about this difference will massively improve the clarity of your

paper. I also take the opportunity to point out that the ozone layer was still significantly affected by elevated concentrations of ODSs in 1980. This is not surprising since ODS concentrations were well above natural levels at this time. I would therefore far rather see reference made to pre-1960 levels since prior to 1960 ODS concentrations were close to or at natural levels.

Yes, for clarity, it will be definitely better to use the word increase in this context. We have put it here instead of "recovery". Regarding the return level, we could write "return to the natural pre-1960 level".

Line 31: I really don't like this phrase "contribute positively to ozone recovery". If you mean recovery of ozone from the effects of ODSs (if not please let me know what influence you're referring to when you speak of its recovery) then of course it goes without saying that $CO_2$ and $CH_4$ cannot contribute to the recovery of ozone from the effects of ODSs. Well, actually, maybe that's not true. If, for example, $CO_2$ or $CH_4$ made ozone less susceptible to the effects of ODSs, then it would be appropriate to say that $CO_2$ and $CH_4$ "contribute positively to ozone recovery". But I don't think that's what you mean here. I think what you mean to say is "It should be noted that not all greenhouse gasses (like $CO_2$ or $CH_4$) contribute to increases in ozone concentrations". And even then you should state how $CO_2$ and $CH_4$ *might* contribute to increases in ozone concentrations.

Agreed. Here, we certainly mean the increase in ozone concentration. So, we corrected the text accordingly. Also, we've moved this sentence down a bit, where we discuss the effect of greenhouse gases on ozone evolution, since it will be better place for this sentence.

Line 32: I strongly disagree that $N_2O$ will "slow down ozone recovery in the future" (assuming by "recovery" you mean recovery from the effects of ODSs), unless you count $N_2O$ as an ODS. Do you? There is, however, significant evidence that increasing atmospheric concentrations of $N_2O$ will delay the return of ozone to pre-1960 levels. But that's a different issue than ozone recovery from the effects of ODSs.

Yes, we totally agree and meant here that the increase of ozone will be slowed down with the $N_2O$ increase. However, in the text we decided to rewrite it based on what you suggested: "$N_2O$ will delay the return of ozone to pre-1960 levels".

Line 33: This use of the term "super recovery" shows the pitfall of conflating ozone increases with recovery of ozone from the effects of ODSs. There is no such thing as super recovery. Ozone, one day, will have fully recovered from the effects of ODSs, i.e., anthropogenic ODSs will no longer have a material effect on ozone. At such a time might ozone concentrations, in some regions of the atmosphere, be higher than prior to 1960? Certainly, most likely because GHG-induced cooling of the stratosphere pushes the odd-oxygen equilibrium towards $O_3$. Is this somehow a "super recovery" from the effects of ODSs. Of course not. It is just that another influence, in addition to ODSs, has affected ozone.

Agreed. In fact, the broadly used in the community term "ozone supper recovery" actually implies that the ozone concentration in some stratospheric regions might be well above that before 1960s. Here, we have reformulated this sentence as follows: "Nonetheless, ozone

concentrations are expected to exceed the pre-ozone hole level in the mid- and high-latitudes because of the increasing greenhouse gas concentrations."

Line 38: In regards to the phrase "In the troposphere, the ozone concentration has been continuously increasing": So would you say that tropospheric ozone has been "recovering" continuously over this period? If not, why not? You seem to refer to increases in stratospheric ozone as "recovery", why not also for tropospheric ozone? Well, for good reason - as you say "due to the continuous increase in tropospheric ozone precursors". Here is your all important *attribution* statement. All I am asking you to do is apply the same reasoning in the stratosphere. When something other than ODSs causes ozone to increase, please don't call this "recovery", just like you don't do so in the troposphere.

We agree with your arguments. We went through the text and checked whether the word "recovery" is used correctly, and if not, then we replaced it with a proper one.

Line 47: Yes, I agree, ozone has been steadily recovering in the upper stratosphere since the late 1990s - attribution studies have shown, unambiguously, that the increases in ozone the the upper stratosphere can be attributed to decreases in Cly and (less so) Bry. This attribution is essential to call "recovery".

Agreed.

Line 47: Do you mean that the recovery is statistically robust or do you mean that the positive trends in ozone in this region of the stratosphere are statistically significantly different from zero at the 2 sigma level? They are two very different things. If you say "This recovery is statistically robust..." it says to me that there is a statistically clear positive influence on ozone resulting from the decline in ODS concentrations. Ozone could still be declining as a result of other factors, but there is a clear positive effect on ozone as a result of declining ODS concentrations. Is this what you mean? Or do you mean that the positive trends in ozone in this region of the stratosphere are statistically different from zero at the 2 sigma level?

Here, we meant that the positive trends in ozone in the upper stratosphere are statistically different from zero at the 2 sigma level. So, for the sake of clarity, this sentence has been rephrased.

Line 49: Your sentence that "The main driver of the upper stratospheric ozone recovery is the reduction of halogen loading" goes without saying if, by "recovery" you mean recovery from the effects of ODSs (if by "recovery" you mean recovery from something other than ODSs, please let me know what it is that your ozone is recovering from). After all, what else could it then be? It would be much more correct to say "The main driver of the upper stratospheric ozone increases is the reduction of halogen loading" because that now opens the door for other factors to be at play which is what you discuss in the very next sentence.

Thank you, we agree, it would be better to say it like this. So, we replaced our sentence with that your suggested.

Line 52: See, here is a great case where you have used the word recovery correctly. See my comment on Line 31. Now here you do indeed show how CH4 can actually promote the recovery of ozone *from the effects of ODSs* because the CH4 promotes the conversion of the Cl radical to the HCl reservoir species. I strongly encourage you to clarify your thinking around "ozone recovery" and "ozone increases". It will make both your paper and your life much clearer.

We agree.

Ljne 54: Regarding "Lower stratospheric ozone (LSO) has recovered more slowly than expected, if at all". I strongly disagree. I believe that the Montreal Protocol has been as effective in reducing the concentrations of Cly and Bry in the lower stratosphere as it has in the upper stratosphere and that, as a result, ozone has been recovering from the effects of ODSs just as well in the lower stratosphere as in the upper stratosphere. It is just that, in the lower stratosphere, factors other than ODSs have been at play that have offset the ozone increases induced by decreases in ODS concentrations such that trends may be statistically indistinguishable from zero, or maybe even negative. But that certainly doesn't mean that ozone is not recovering from the effects of ODSs in the lower stratosphere.

We agree with your comment here, it is definitely better to use the word "increase" than "recovery", since we did not mean that ozone is not recovering in the lower stratosphere from the effects of hODS. So, we replaced it here.

Line 69: Please include references to support the assertion that there have been changes in the relative strengths of the lower and upper branches of the BDC (and please expand this acronym) and that these changes have had an effect on ozone.

Agree. We updated this sentence as follows: «Besides a change in the relative strengths of the lower and upper branches of the meridional Brewer-Dobson circulation (BDC, Butchart et al., 2006) resulted in ozone changes (Keeble et al., 2017; Le et al., 2009; Oman et al., 2009)....»

Line 73: Unlike the other factors listed, "insufficient treatment of diffusion and transport processes in models" definitely does not cause a decline in LSO. It may cause models to incorrectly simulate the decline in LSO but it does not cause a decline in LSO in the same way that recent reductions in solar activity might.

Yes, it is right. We decided to exclude this part from this paragraph and move it to the paragraph below where we discuss problems in models preventing the accurate reproduction ozone trends in LSO.

Line 77: The phrase "pattern of the signal in LSO" is a little vague. Do you you mean the pattern of trends in LSO as a function of latitude (and maybe also altitude)?

Yes, here we meant the pattern as a function of latitude and pressure levels. So, we added these words to the sentence.

Lines 78-79: Replace "the persistent" with "the observed persistent" just to make it clear that these negative trends are seen in observations.

Done.

Line 84: I would have expected the exact opposite to be true, i.e., if you have a large ensemble of simulations, by taking the ensemble average you smooth out all of the unforced variability leaving you only with the forced variability which is the signal that you're after. Ah, but maybe you mean this: Over the past 60 years say, ozone has evolved within the context of a very specific sequence of unforced dynamical events, i.e., events driven by the chaotic nature of the climate system. That sequence could be quite anomalous (extreme). We don't know because we have only one planet with a single trajectory through history. It could be, but we have no way of telling, that around the time when Thomas Peter was born, if there had been some perturbation to the state of the atmosphere that the evolution of dynamical events that has affected LSO may have been completely different and LSO would have actually increased. But with a single planet with a single history, we can't say. Ah, but we can to some extent. Taking ESMs as a proxy for our climate system we can see if *any* single ensemble member has a LSO trend that looks like what we have seen other the past 60 years. If there is one, we can say "Ah ha, that's the timeline that our planet followed to the exclusion of all others". It is a plausible (though quite different from the mean) timeline. But if you had 1,000 ensemble members and none of them had a LSO trend such as that observed, then you would conclude either that (1) what we have experienced on this planet in the last 60 years in terms of the sequence of dynamical events that have affected LSO is highly anomalous, or (2) that there is something wrong with the models. So this is how I could see that looking at individual models could be useful in this regard. But I don't feel that that's what you communicated.

Yes, we (and Stone et al. 2018) meant that the model might reproduce the ozone trends in the lower stratosphere, if the model dynamics at least in one ensemble member resembles the dynamical pattern that happened in reality. Increasing the number of ensemble members to 100 or 1000 will increase the probability to get this pattern in the model atmosphere which would allow obtaining ozone trends as closely as possible to the observed ones. Such complication comes from the fact that the considered period is rather short and therefore such small, not properly resolved, dynamical fluctuations cause a very strong effect on the derivability of the trends. We cannot perform 1000 simulations, due to obvious computational limitations, however, the obtained range of results for this region in our and Stone et al. (2018) work already allows us to state that internal variability is a strong player in this game and its specific combination can produce various "technically" statistically significant results.

Stone, K.A., Solomon,S., and Kinnison, D. E.: On the Identification of Ozone Recovery, , 45, 5158–5165, https://doi.org/10.1029/2018GL077955, 2018.

Here, we rephrased this part as follows:

"These results suggest that for the validation of the past long-term ozone trends in the dynamically controlled regions, it is essential to have many ensemble members and analyze

these experiments separately to establish whether there is any member reproducing observed trends better."

Line 89-90: Ah yes, here comes the nail in the coffin of the ESMs - even with assimilated dynamics the ESMs can't reproduce the LSO trends. As a result, I was then surprised that your paper concludes with the statement that "the obtained qualitative agreement between the model and observations allows to produce reliable estimates of future ozone evolution using modern chemistry-climate models". I think that conclusion is not support b y your analysis.

Yes, from one side SOCOLv4 ESM as well as its predecessors cannot fully reproduce the distribution and the magnitude of observed ozone trends in the lower stratosphere probably for the reason we discussed above. However, we see that in some ensemble members (4 and 5) LSO trends much resemble the observed ones. In other atmospheric regions, the model reproduces well the trends in ozone against observations. So, we could say that SOCOLv4 ESM is able to reproduce the LSO trends if unforced variability would fit, by chance, the pattern we have in reality. So, the model generally reproduces trends in the atmospheric ozone with enough quality to use this model to predict its future evolution. Moreover, the trends from satellite data are not as robust as we would like to have. Short time series, linear regression analysis, quality/completeness of the applied proxies and strong interannual variability (e.g., Chipperfield et al, 2018) suggest that the obtained "observed" LSO trends have some unknown uncertainty level which is hard to quantify.

Moreover, assimilated dynamics does not make a CTM from an ESM, i.e., there is still a lot of freedom for the model dynamics to act. A recent CCMI-1 analysis, for example, showed that the inter-model spread of nudged simulations in the stratosphere can result in a similar or even larger spread than in free-running simulations (Chrysanthou et al., 2019).

Chipperfield, M. P., Dhomse, S., Hossaini, R., Feng, W., Santee, M. L., Weber, M., Burrows, J. P., Wild, J. D., Loyola, D., and Coldewey-Egbers, M.: On the Cause of Recent Variations in Lower Stratospheric Ozone, , 45, 5718–5726, https://doi.org/10.1029/2018GL078071,2018.

Chrysanthou, A., Maycock, A. C., Chipperfield, M. P., Dhomse, S., Garny, H., Kinnison, D., Akiyoshi, H., Deushi, M., Garcia, R. R., Jöckel, P., Kirner, O., Pitari, G., Plummer, D. A., Revell, L., Rozanov, E., Stenke, A., Tanaka, T. Y., Visioni, D., and Yamashita, Y.: The effect of atmospheric nudging on the stratospheric residual circulation in chemistry–climate models, Atmos. Chem. Phys., 19, 11559–11586, https://doi.org/10.5194/acp-19-11559-2019, 2019.

Line 95: No I am pretty sure that this is not what Avallone and Prather argued - the first paper to describe why tropical LSO might decrease. Rather, the more rapid vertical motion in the tropical troposphere and lower stratosphere results in there being less time for photochemical ozone production in the rising air parcels and hence a reduction in LSO. Please read Avallone, L.M., and M.J. Prather, Photochemical evolution of ozone in the lower tropical stratosphere, Journal, of Geophysical Research, 101(D1), 1457-1461, 1996.

Thank you, this is an interesting concept and reference. Indeed, we should be more precise in explaining the reason for the tropical LSO decline here. So, we added this citation and reformulated the sentence in order to describe this process as follows:

"An intensification of air parcel rising associated with an acceleration of the meridional Brewer-Dobson circulation (Butchart et al., 2006) that causes less time for photochemical ozone production, was suggested as the primary mechanism for the declining ozone trend in tropical LSO (Avallone and Prather 1996)."

Line 164: Presumably these are zonal winds *at the equator*. You should state that.

Yes, it is. It has been added.

Figure 1: It is hard to distinguish the results. I think that either tabulating these results or, if you want to retain the graphical format, slightly horizontally offsetting the data points from each other would help a lot.

The reason to have this figure here is to show that the correlation coefficients are low enough to consider these proxies usable for the regression analysis. We agree that it would be good to have more space between these points to make them more readable. We replotted this figure as follows:

[Figure]

Line 182: Why not just apply Gram-Schmidt orthogonalisation (https://en.wikipedia.org/wiki/Gram%E2%80%93Schmidt_process) to ensure orthogonality across your basis functions?

Yes, this method can also be used. However, in our study, we found it reasonable to use the standard set of proxies (to be compatible with previous publications) and a simple correlation test to clearly visualize what the mutual correlation between proxies is.

Line 185: This is the first mention of "background level of the model". I think you need to explain in one additional sentence exactly what this is.

Indeed, it is important to do so. We reformulated this sentence like this: "The evolution of the predicted variable is characterized in DLM by the so-called "trend-term" or background

level. Straightforwardly, it is the evolution of a variable filtered from the known part of its variability, induced by external forcings, which are represented in DLM by proxies."

Lines 194-195: Yes, exactly. Without such attribution you cannot speak of any ozone recovery from the effects of ODSs. If you found that ozone was increasing but also found that that ClOx and BrOx destruction cycles were strengthening, you would not call that increase "ozone recovery". So I am pleased to see this recognition that discussion of recovery must be done in the context of this attribution to changes in the strengths of the ClOx and BrOx cycles.

We agree and have corrected it throughout the text.

Line 214: When you refer to upwelling here I assume you are referring to atmospheric upwelling and not oceanic upwelling, but it may be worth clarifying that.

Yes, here, we meant the atmospheric upwelling. We added this to the text.

Line 219: I wouldn't call 55°S-55°N "near-global". I think that it would be more accurate to refer to this as "extra-polar". It also says that these are monthly mean anomalies but they look more like annual mean anomalies to me.

Yes, we agree, it can be called extra-polar. Also, you are right in guessing that the figure shows annual mean evolutions, but here in the text it was wrongly written "monthly mean". We have corrected both inaccuracies in the text.

Line 224: Unless the increase in mesospheric ozone can be attributed to a decrease in Cly and/or Bry I would disagree that "mesospheric ozone has a tiny positive contribution to the recovery of the total column ozone on the near-global scale". It may contribute to increases in total column ozone but, with attribution to Cly and Bry, it cannot be said to be contributing to the recovery of ozone from the effects of ODSs.

Yes, here, we meant that the increase of mesospheric ozone slightly contributes to the overall increase of total column ozone.

Line 227: With regard to "started to recover distinctly". I believe it would be more correct to say "started to increase distinctly" because you have not demonstrated conclusively that the increase is a result of a decline in Cly and Bry.

Yes, it is better to phrase this part like this: "…started to increase distinctly".

Line 228: See here you refer to "pronounced increase in ozone" but don't call it "recovery". Why not? Elsewhere where there has been an increase in ozone you call it "recovery" so why not here? Can you see the danger of referring to "increasing ozone" as "recovering ozone" when you don't actually do the attribution to decreasing concentrations of Cly and Bry?

We agree that it was incorrect wording in this context. We have corrected it in the text.

Line 233: Do you mean "ozone recovery" or "ozone increases"? And you need to say much more about this "bias shift" in the ERA5 reanalyses and what causes it.

Here, we rewrote it like this: "In ERA-5, the start of ozone recovery from the effects of hODSs is biased relative to the BASIC and SOCOLv4 towards the beginning of the 2000s."

Line 235: So if the minimum occurred in 1992, would you say that ozone was then recovering in this region of the stratosphere from 1992 onwards?

Yes, we can say that the minimum of ozone concentration is observed for a few years after the Pinatubo eruption, and then, after the ozone starts to recover from the effects of hODSs.

Line 239: Highly perturbed by what? Or do you just mean is highly variable just as a result of natural unforced variability.

Yes, it is better to say that the ozone is highly variable as a result of natural unforced variability.

Line 244: I really don't know what you mean here by "essential artifacts from unknown origins". This needs to be explained much more carefully.

This is a good question. Actually, in figure 3, you can see that the behavior of the ozone evolutions from the MERRA-2 and ERA-5 are generally very different from that in BASIC and SOCOLv4, especially in the stratosphere. The reason for such a behavior is not related to DLM as the DLM fits and "clear" evolutions (time series minus climatology; dashed line in Figure 3) of ozone from these reanalyses agreed with each other. So, the problem is definitely with these reanalyses data sets as such. However, what is behind this anomalous behavior is still unclear and we just suggested some of the possible reasons. This might be an issue related to the merging of observational data or specific data retrievals. Also, there could be some issues with model simulations used to fill the gaps in observations as models might be with simplified physics, chemistry, and/or dynamics. Hence, our outcome is that these data, at least in this version, have limitations to be used in trend analysis. The MSRv2 reanalysis behavior is much smoother and more understandable, which agrees rather well with the SOCOLv4 simulations and other studies.

So, we reformulated this as follows:

"MERRA-2 in the troposphere is not shown because it dramatically disagrees with expectations and other data sets….. We cannot demonstrate the source of the problem, but it could be related to some issues with model simulations, observation data or assimilation approach."

Line 246: ERA5 shows biases against what?

The ozone evolution from ERA5 is biased against those from BASIC and SOCOLv4.

Line 262: Why only "trend might result from continuous emissions of tropospheric ozone precursors". Can you not be more definitive about this?

Yes, we can, and actually, we did it a bit earlier when we discussed the extra-polar ozone evolutions and drivers for ozone evolution (Figure 2). The positive trend in tropospheric ozone is mostly related to the increase in NOx and other precursors like CO which is the product of CH4 oxidation. So, here, we have updated this sentence a bit to make the reader's life easier as follows: "The positive trend might result from continuous emissions of tropospheric ozone precursors, in the extra-polar region mainly NOx (see Figure 2) and CO…" .

Line 265-266: What vertical region of the atmosphere are you referring to in this sentence?

Here, we refer to the lower stratospheric region.

Line 269: I think that it would be better to say "the expansion of the ozone towards lower latitudes" and more accurate to avoid confusion that you may be referring to vertical expansion.

Yes, it would be better to say like this.

Line 270: With regard to "Most likely". Surely you can easily confirm this by looking at the position of the jet in SOCOLv4 and comparing it to the position of the jet in the reanalyses?

We reformulated this sentence as follows:

"it is related to overestimated isolation of the southern polar vortex area during winter time (Sukhodolov et al., 2021)."

Line 282: See, here you are referring to 1998-2018 as the recovery period and yet, in some regions of the stratosphere, and in some ensemble members, ozone is decreasing. And this is entirely OK and correct because "ozone recovery" does not necessarily mean "ozone increases". There is no doubt in my mind that even in those strong blue regions in Figure 5 that ozone is recovering from the effects of ODSs in those regions. It's just that that recovery is being overwhelmed by some other process.

We agree.

Line 286: "ozone increase in the mesosphere" rather. And please elsewhere in this section be very clear when you are referring to "ozone recovery" (from the effects of ODSs) and more generally "ozone increases". I am not going to highlight every instance where you have conflated the two.

Agree. We went through the text and corrected the places where there is this conflation.

Lines 292-304: I think that this paragraph at the top of page 15 of the paper is the most valuable aspect of the work and more should be made of this (see also by comment regarding line 84). The variability in LSO trends across the different ensemble members is very telling, even if they are not statistically significant. The point here is that the BASIC trends represent just one realisation of our planet's history - and of course we have just one. But the SOCOLv4 members show what, conceivably, could also have happened. What

happened in our single reality may well be a statistical outlier across the full range of histories that conceivably could have happened, but it happened nonetheless. So the message for me is: don't panic if you do an ESM simulation that doesn't show the observed trends in LSO from 1998 to 2018. It may just be that your particular ensemble member didn't follow the same "dynamical storyline" that happened in reality. And don't even think about looking at the ensemble mean - that's not going to help if what happened in our planet's 1998-2018 history is a statistical outlier. A better way to go is to conduct a large ensemble of simulations and identify those simulations which do look at lot like what happened in reality, even if there are relative very few of them. Then diagnose that subset for the underlying causes of the shown LSO trends. No sense in diagnosing ESM simulations where, for reasons of different unforced natural variability, the ESM's "dynamical storyline" is different from what happened in reality. Take a look at "North Atlantic climate far more predictable than models imply" by Smith, D.M.; Scaife, A.A.; Eade, R.; Athanasiadis, P.; Bellucci, A., et al., Nature. They did something similar. I think that what you have done here is important in this regard and something worth exploring further - not necessarily in this paper but perhaps in a follow-up paper.

Yes, we totally agree with your arguments. We have added these ideas to section 5.

Added text:

"We show that the model cannot fully reproduce the observed trends in LSO from 1998 to 2018. It may just be that our particular ensemble members do not follow the same dynamical trajectory that happened in reality. Therefore, the use of ensemble mean is not going to help if what happened in Earth's atmosphere 1998-2018 history is a statistical outlier. A better way to proceed is to conduct a larger ensemble of simulations and identify those simulations which do look a lot like what happened in reality, even if there are relatively very few of them. Then diagnose that subset for the underlying causes of the shown LSO trends, as was done by Smith et al. (2020). Our analysis of six individual realizations for the ozone recovery phase revealed that the LSO trend patterns in ensemble members 4 and 5 show a better resemblance with observations than other members of the ensemble, suggesting that the natural variability in simulations may coincide, by chance, with the dynamical storyline that we have in our reality i.e. in observations. "

Line 317: With regard to "that the modeled ozone changes in the extratropical lower stratosphere are still uncertain", I would say that it's more the case that your simulations show significant variability in these trends across ensemble members suggesting that unforced natural variability can create trends in LSO that are not that different from what is observed. We don't know how LSO in reality could have evolved under slightly different initial conditions since we have only one planet with a single history. But SOCOLv4, as a proxy, does show that very different LSO trends can result from different natural dynamical variability.

Yes, here we agree with your statements and found it better to rewrite this paragraph to make the results more valuable based on your suggestion as follows:

"Although ozone trends in SOCOLv4 are well simulated in most atmospheric layers, our study infers that the modeled ozone trends in the extra-polar lower stratosphere show

significant variability in these trends across ensemble members. Indeed, detecting clear and robust ozone trends outside the polar regions in model simulations is difficult because of the high dynamical variability perturbing the individual ensemble members (Stone et al., 2018). This justifies analyzing each ensemble member separately to study extra-polar LSO variability, as we do in the present work…. " (see the response for previous comment too)

Line 321: While I agree that "no single member (amongst the six) suggests an extensive negative signal found in the BASIC", given the variability you see in your six members, what do you think the likelihood would be of finding a single member that closely matches reality if you had a 1,000 member ensemble? My guess would be "fairly high". If you found one (and you really only need one - you could argue that that's the particular "dynamical storyline" our planet followed) and diagnosed that single run for the drivers of the LSO trends, I think that this could provide valuable insights into the drivers of the observed trends.

Yes, it is worth doing in the future when and if such trend patterns are identified in the simulations using similar methodology as Smith et al. (2020).

Line 326: I agree with the statement that "Yet, the other four members have less pronounced but still negative trend regions", but who cares? Those are simulations with different dynamical storylines to what our planet experienced and so can be discarded. Just diagnose number 4 and 5. And definitely don't waste time analysing the ensemble mean.

Agree, we reformulated this part in a way to point the attention to rather interesting single ensemble members than the ensemble mean or trend distributions from realizations with less agreement with observations.

Line 334: I agree that "ozone recovery period is not yet long enough to detect statistically significant LSO trends in the LOTUS regression analysis" but if you could run, e.g., ensemble 4 with detailed tracking of the contribution of different chemical cycles to the ozone tendencies in SOCOLv4, e.g. as was done in Revell, L.E.; Bodeker, G.E.; Huck, P.E. and Williamson, B.E., Impacts of the production and consumption of biofuels on stratospheric ozone, Geophys. Res. Lett., doi:10.1029/2012GL051546, 2012 and Revell, L.E.; Bodeker, G.E.; Huck, P.E.; Williamson, B.E. and Rozanov, E., The sensitivity of stratospheric ozone changes through the 21st century to N2O and CH4, Atmos. Chem. Phys., doi:10.5194/acp-12-11309-2012, 2012, then you could probably say something really new and novel about the recovery of LSO *from the effects of ODSs* over the 1998-2018 period.

Thanks for the advice. We will try to do it in the future.

Line 352: With regard to "does not yield the observed negative trends", yes but you only had 6 members. I bet if you had 1,000 members you would find one that does track reality well. But that would be well beyond the compute budget for ETH.

Yes, we decided (as it was mentioned before) to rephrase the discussion so that readers do not think that we want to say that the model cannot at all reproduce the pattern of trends from reality. In our study, despite only 6 members of the ensemble, some of the trend

patterns in LSO were able to get close to what was formed in our reality. And as you correctly noted, if you make 1000 members of the ensemble, the probability of getting what is observed will definitely increase. As such, we have rephrased the discussion by emphasizing that the model can, under certain conditions, reproduce the observed LSO trends.

GRAMMAR AND TYPOGRAPHICAL ERRORS

There were  grammatical errors in the paper which I did not take the time to correct. I would strongly suggest that the authors find someone to thoroughly proof read the paper. Please make sure that all acronyms are expanded at the place of first occurrence.

The text proofreading has been done.

Line 27: Small suggested change in wording from "total ozone is expected to recover" to "total column ozone has been expected to recover"

Done.

Line 42: The more common phrase is "stratosphere-troposphere exchange" but this is not a big deal so leave it as it is of you wish.

Ok. It has been corrected.

Line 82: Replace "dynamic variability" with "dynamical variability".

Done.

Line 179: Previously this was referred to as the "ozone depletion phase", now referred to as the "ozone-depleting period".

We replaced "ozone-depleting period" with "ozone depletion phase" within the text in sake of homogeneity of notations.

Line 184: Replace "output DLM output" with "DLM output".

Done.